# Unambiguous Models and Machine Learning Strategies for Anomalous Extreme Events in Turbulent Dynamical System

**DOI:** 10.3390/e26060522

**Published:** 2024-06-17

**Authors:** Di Qi

**Affiliations:** Department of Mathematics, Purdue University, 150 North University Street, West Lafayette, IN 47907, USA; qidi@purdue.edu

**Keywords:** turbulent systems, machine learning, multiscale modeling, long short-term memory

## Abstract

Data-driven modeling methods are studied for turbulent dynamical systems with extreme events under an unambiguous model framework. New neural network architectures are proposed to effectively learn the key dynamical mechanisms including the multiscale coupling and strong instability, and gain robust skill for long-time prediction resistive to the accumulated model errors from the data-driven approximation. The machine learning model overcomes the inherent limitations in traditional long short-time memory networks by exploiting a conditional Gaussian structure informed of the essential physical dynamics. The model performance is demonstrated under a prototype model from idealized geophysical flow and passive tracers, which exhibits analytical solutions with representative statistical features. Many attractive properties are found in the trained model in recovering the hidden dynamics using a limited dataset and sparse observation time, showing uniformly high skill with persistent numerical stability in predicting both the trajectory and statistical solutions among different statistical regimes away from the training regime. The model framework is promising to be applied to a wider class of turbulent systems with complex structures.

## 1. Introduction

Extreme events and the related anomalous statistics are fascinating phenomena universally observed in a wide class of natural and engineering systems [1,2,3,4,5]. An active, contemporary topic with a grand challenge is understanding, predicting, and controlling such events using qualitative and quantitative models [6,7,8,9,10]. Dynamical systems with extreme events are often characterized by strong internal instabilities and the competing effects of coherent large-scale structures and multiple interacting small-scale processes [11,12,13]. The accurate quantification of such features requires solving complex nonlinear equations among different parameter regimes to draw a complete picture of the statistical solution profile. Direct strategies by explicitly resolving all the scales with many repeated evaluations become inefficient and often impractical due to the very high computational overload [14,15]. Effective modeling and parameterization methods are still needed to capture the key dynamical features with computational efficiency and robustness to the noise errors amplified by inherent instabilities.

Data-driven modeling using machine learning ideas [16,17,18,19] has become one appealing approach to learn the unresolved physical processes given sufficient data covering complete solution regimes. Such data-driven strategies have shown potential in recovering unresolved subscale dynamics which are difficult to derive via first principles [20,21,22], or suffer high computational cost in direct approaches [23,24,25,26,27]. The increasing amount of observational data further helps the development of various data-driven models to advance the understanding of the underlying physical mechanisms and thus to provide fast and accurate solvers [28,29,30,31,32]. In the case of learning model dynamics showing extreme events and anomalous statistics, however, the available data for training are often restricted with incomplete observations (such as a limited dynamical regime and sparse measurements) and are polluted with various model errors (such as amplified noise and uncertainties from the model instability, as well as the imperfect model approximation). The challenge remains to find a universally applicable model framework with uniform prediction skill among different statistical regimes that are beyond the limited training dataset. Large model uncertainty and exponentially growing model error often lead to the breakdown of long-time prediction typical for complex turbulent systems without a complete physical understanding. An effective model of turbulent dynamical systems requires computational stability against noises and perturbations.

In this paper, we aim to investigate useful machine learning techniques to recover the unresolved components of complex physical systems with coupled multiscale processes. In the development of new machine learning strategies, it is useful to start with simplified prototype models, where the key dynamical structures of interests can be identified for a thorough understanding. We propose a group of unambiguous models for both the understanding of key dynamical structures in generating the representative extreme phenomena, and also the development of effective machine learning strategies based on explicit physical structures. The unambiguous models are drawn from geophysical turbulence [33,34,35], and accept analytically tractable conditional Gaussian structures incorporating essential processes of multiscale flow interaction and the turbulent transport of passive tracers. The system generates representative anisotropic flows with topographic blocking, qualitatively resembling those observed in the midlatitude ocean and atmosphere [36,37,38], and passive tracer fields exhibiting strong extreme events with skewed or fat-tailed probability distributions comparable to laboratory experiments [6,29,37,39]. In addition, the model fits into the general conditional Gaussian framework [32,40,41,42], which provides explicit analytic formulas by expressing the small-scale variables as a group of Gaussian processes depending on the realizations of the observed large-scale process. The explicit analytical formulas draw a detailed characterization of the trajectory solution structures as well as the anomalous statistics. Based on the explicit solutions of the detailed large–small-scale interaction mechanism, effective machine learning methods are developed, only replacing the unresolved small-scale processes by data-driven models. Such simple but comprehensive equations are shown to serve as a group of unambiguous test models for the central problems in the development of data-driven strategies with various datasets and model architectures.

### Contributions of This Work

We introduce novel modeling strategies that are able to learn the hidden dynamical processes coupled with multiple temporal and spatial scales. The neural network model can be trained under a single set of data with sparse time measurements, and the trained model is capable of predicting different types of extreme events and distinct statistical features among various dynamical regimes where data are unavailable. By construction, the new neural network architectures provide an accurate higher-order approximation of the original dynamics informed of its essential physical structures. The coupled multiscale processes are modeled efficiently by exploiting the conditional Gaussian structure and decomposing the full model into smaller subsystems that are easier to be learned. In addition, the model stability for long-time iterations is effectively improved through training with a feasible loss function considering multistep outputs, including model errors.

In building the neural network model, the original long short-term memory (LSTM) network [43,44] is improved by adding detailed inner connections to track the correlated long-time history in the data from turbulent signals. The new neural networks are applied for multiple small scales for efficient modeling, and the outputs for different scales are combined in the explicit large-scale mean flow equation to introduce physics-based updates to the learning process. We improve the idea in [31,32] to introduce the loss function for the optimization procedure by combining the use of a new relative entropy distance [45,46] and the standard mean square error. The calibration of model approximation errors is made to focus on the dominant shapes of the turbulent signals instead of an exhausting fitting of the unnecessary pointwise turbulent errors. With the combined contributions of the new designs of model architectures and loss functions, the proposed neural network model overcomes the inherent difficulties of early divergence and large training errors common among the traditional LSTM networks [17,47].

The neural network model is then tested on the proposed unambiguous model from geophysical turbulence with the large mean flow interacting with two small-scale modes. The model provides the simplest setup restricted to a two-mode interaction, while still maintaining a variety of dynamical regimes displaying different types of extreme events for testing the skill of the neural network. The neural network model focusing on the small-scale processes is trained in one statistical regime from the available partial data. Then, the trained model can be applied universally for various scenarios with distinct non-Gaussian statistics. By applying the model to different datasets in the prediction stage, we show the predictive capability in the trained model to recover the key dynamics from incomplete data and limited information. The model also allows a sparse dataset with longer time measurement steps, showing stable performance for long-time prediction. The model with unresolved processes can be further generalized to a wider class of complex turbulent systems [1,48,49] to construct computationally efficient, reduced models with nonlinear high-order feedback [26,27].

In the rest part of the paper, the unambiguous model framework with explicit solutions and representative statistical regimes is introduced in Section 2. The general machine learning strategy to learn the complex dynamical processes is constructed in Section 3. Then, the neural network model is combined with the explicit physical structures in the dynamical system in Section 4 to capture extreme events and the statistical features. Numerical tests are carried out in a two-mode topographic model in Section 5 as an illustration for the scope of skill of the strategy. A summary with discussions for future research directions is given in Section 6.

## 2. An Unambiguous Model Framework for the Investigation of Extreme Events

We first propose a group of prototype models with tractable mathematical structures to serve as a clean testbed for the investigation of the various distinctive phenomena found in natural systems. The models are constructed by including the key features in realistic turbulent systems, such as a wide variety of extreme events and anomalous statistics with fat-tailed or skewed probability density functions (PDFs). The prior information of the dynamical system is then exploited for guidelines to design new neural network architectures in the next sections.

### 2.1. General Formulation of the Unambiguous Mathematical Models

A general mathematical framework for a wide group of systems can be introduced in the following abstract form for the multiscale states $u=\left(u_{1},u_{2}\right)\in R^{N_{1}+N_{2}}$
(1)$$\begin{array}{cc} du_{1}= & \left[A_{0}\left(t,u_{1}\right)+A_{1}\left(t,u_{1}\right)u_{2}\right]dt+\Sigma_{1}\left(t,u_{1}\right)dW_{1},\\ du_{2}= & \left[B_{0}\left(t,u_{1}\right)+B_{1}\left(t,u_{1}\right)u_{2}\right]dt+\Sigma_{2}\left(t,u_{1}\right)dW_{2}. \end{array}$$

Usually, $u_{1}\in R^{N_{1}}$ can be viewed as the observed slow process and $u_{2}\in R^{N_{2}}$ as the unobserved fast states in a much larger phase space $N_{2}\gg N_{1}$. A conditional Gaussian framework [42,50] is developed based on the general formulation (Equation 1), where $u_{2}$ can be expressed as a Gaussian process given the complete history of $u_{1}\left(s\right){|}_{s\leq t}$
$$p\left(u_{2}\left(t\right)|u_{1}\left(\cdot \right)\right)=N\left({\bar{u}}_{2}\left(t;u_{1}\left(\cdot \right)\right),R_{2}\left(t;u_{1}\left(\cdot \right)\right)\right),$$
where ${\bar{u}}_{2},R_{2}$ are the conditional mean and covariance matrix depending on the realization of $u_{1}$. This implies that the final probability distribution for the process $u_{2}$ can be expressed as a mixture conditional on different realizations of the ‘observables’ $u_{1}$. Still, the full probability distribution of the system (Equation 1) could be highly non-Gaussian as a combination of all the probability realizations from the above conditional Gaussian structure
$$p\left(u_{2},t\right)∼\int p\left(u_{2}\left(t\right)|u_{1}\right)d\mu \left(u_{1}\right),$$
where $\mu \left(u_{1}\right)$ is the probability measure for the entire observed process $u_{1}\left(s\right),s\leq t$. As a further comment, many realistic turbulent systems such as those found in climate forecast may not exactly follow the conditional Gaussian structure (Equation 1). Therefore, usually, additional approximations are needed with the introduction of imperfect model errors.

#### Topographic Barotropic Model with Large-Scale Mean Flow Interaction and Strong Small-Scale Feedbacks

We focus on a special group of the general model framework (Equation 1) with reference to geophysical turbulent flows. The topographic barotropic system [33,34] models the complex interactions of a large-scale mean flow *U* and small-scale vortical fluctuations *q* in quasi-geostrophic turbulence
(2a)$$\begin{array}{ccc} \frac{\partial q}{\partial t}+v\cdot \nabla q & = & D\left(\Delta \right)\psi +F_{q}, \end{array}$$
(2b)$$\begin{array}{ccc} \frac{dU}{dt}+⨏\frac{\partial h}{\partial x}\psi^{\prime } & = & -d_{U}U+F_{U}. \end{array}$$

The topographic model is defined on the two-dimensional doubly periodic plane $x\in D={\left[-\pi ,\pi \right]}^{2}$ for simplicity, with the potential vorticity *q*, stream function ψ, and flow velocity v defined as
(3)$$q=\nabla^{2}\psi^{\prime }+h+\beta y,\;\psi =-U\left(t\right)y+\psi^{\prime },\;v=\nabla^{⊥}\psi =\left(U-\partial_{y}\psi^{\prime },\partial_{x}\psi^{\prime }\right).$$

Above, the small-scale stream function $\psi^{\prime }$ is separated from other large-scale terms. There exists a multiscale coupling between the small-scale fluctuations ([Disp-formula FD2a-entropy-26-00522]) and the large-scale uniform mean flow (2b) through the domain-averaged quantity $⨏\frac{\partial h}{\partial x}\psi^{\prime }\equiv \frac{1}{\left|D\right|}\int_{D}\frac{\partial h}{\partial x}\psi^{\prime }$ with $\left|D\right|$, the computational domain area. The model (2) combines several crucial features in geophysical turbulence [1,35]: the effects of topography (*h*), rotation (β), external forcing ($F_{q},F_{U}$), frictions $D\left(\Delta \right)$ for the fluctuation states (for example, Ekman drag *r* and higher-order dissipation $\nu \Delta^{2}$) and linear damping $d_{U}$ acting on the large-scale mean flow *U*. It is easy to check that the model fits into the general framework (Equation 1) by setting $u_{1}=U$ as the large-scale mean flow and $u_{2}=q$ for all the small-scale processes.

Using the flow solution of (2), we can introduce an additional equation modeling the turbulent transport of passive tracers through the advection and diffusion of the passive tracer density field $T\left(x,t\right)$
(4)$$\frac{\partial T}{\partial t}+v\cdot \nabla T=-d_{T}T+\kappa \Delta T,$$
where the advection flow $v=\nabla^{⊥}\psi$ is provided by the velocity solution in (Equation 3), and the tracer field is subject to damping and diffusion effects due to parameters $d_{T}$ and κ on the right-hand side. A wide variety of properties are found and analyzed under this tracer framework (Equation 4) for both theories and applications in turbulent transport and diffusion [6,29,51]. Again, the tracer Equation (Equation 4) can be also categorized into the general framework (Equation 1) with $u_{2}=T$ and no direct feedback to the flow state $u_{1}=v$. Instabilities and uncertainties are introduced through the multiscale interactions [35,52] in the above flow system (2) as well as the passive tracer field (Equation 4). The prediction of both the large and small processes from the partially observed data and unknown dynamics forms a general challenge for the accuracy and stability.

Despite the simplicity, the stringent paradigm models for multiscale coupled flow (2) and passive tracer (Equation 4) fields exhibit complex statistics characterizing a number of crucial realistic phenomena, such as the atmospheric blocking, topographic instability, and nonlinear energy transfer through scales [1,33]. The tractable dynamical structures in the simplified models enable a series of detailed analyses for the mathematical understanding of various physical processes [24,35] and the construction of comprehensive computational strategies [29,34,53].

### 2.2. Analytical Solutions from the Conditional Gaussian Models

Still, the model ([Disp-formula FD2a-entropy-26-00522]) couples all the small-scale fluctuation modes through the nonlinear advection term on the left-hand side. This will lead to very complex dynamical structures for the analysis. Here, in order to find more analytically tractable solutions, we propose further simplification to the original system so that we are able to focus on the most important mean–fluctuation interactions in determining the final flow structure.

Starting from the topographic barotropic model (2) and the corresponding passive tracer Equation (Equation 4) on the periodic domain *D*, we project the states of the flow velocity field $v=\partial_{x}\psi$ and the tracer density field *T* on the Fourier spectral space by
(5)$$v\left(x,t\right)=\sum_{k}{\hat{v}}_{k}\left(t\right)e^{ikl\cdot x},\;T\left(x,t\right)=\sum_{k}{\hat{T}}_{k}\left(t\right)e^{ikl\cdot x}+\alpha y.$$

Above, the *layered topography* [29] is applied along one characteristic wavenumber direction l with $\left|l\right|=1$ (for simplicity, we take $l_{x}=1,l_{y}=0$ in the following analytical results without loss of generality). In addition, a background mean gradient profile αy along the *y* direction is assumed for the tracer density field. The layered topography eliminates the nonlinear interactions between the fluctuation modes, thus enabling us to focus on the coupling effect between the large and small scales. Then, we aim to find the trajectory and statistical solutions of the following coupled flow system under the spectral representation
(6a)$$\begin{array}{cc} \frac{d{\hat{v}}_{k}}{dt}-i\left(k^{-1}\beta -kU\right){\hat{v}}_{k}+{\hat{h}}_{k}U & =-d_{k}{\hat{v}}_{k}+\sigma_{v,k}{\dot{W}}_{k}, \end{array}$$
(6b)$$\begin{array}{cc} \frac{dU}{dt}-\sum_{k}h_{k}^{*}{\hat{v}}_{k} & =-d_{U}U+\sigma_{U}{\dot{W}}_{0}. \end{array}$$

Above, through orthogonal projection (Equation 5), the linear terms are decoupled into each spectral mode ${\hat{v}}_{k}\left(t\right)$. The integration representing topographic stress on the left-hand side of (2b) becomes the summation over all the spectral modes from the inner product. In particular, the nonlinear coupling between small-scale modes in ([Disp-formula FD2a-entropy-26-00522]) vanishes due to the assumed layered topography along one wavenumber direction l. The additional unresolved effects are summarized in the white noise as $F_{q,k}=\sigma_{v,k}{\dot{W}}_{k}$ and $F_{U}=\sigma_{U}{\dot{W}}_{0}$. Accordingly, the associated passive tracer equation is given by
(6c)$$\frac{d{\hat{T}}_{k}}{dt}=\left(-\gamma_{T,k}-ikU\left(t\right)\right){\hat{T}}_{k}-\alpha {\hat{v}}_{k},$$
with $\gamma_{T,k}=d_{T}+\kappa k^{2}$ and white noise amplitudes $\sigma_{v,k},\sigma_{U}$ defined for small and large scales. The shear flow modes ${\hat{v}}_{k}$ serve as a forcing on the passive tracer mode ${\hat{T}}_{k}$ induced by the mean gradient α. The detailed derivation of the equations and their properties are discussed in [6,29,34] with many applications.

In particular, we refer to the large-scale mean flow *U* as the ‘observed state’ that is measured at a time frequency Δt (note that this is usually much longer than the admissible integration time step in the direct numerical scheme); the small-scale velocity modes ${\hat{v}}_{k}$ with important feedback to the mean flow equation are treated as the unresolved states. In the following sections, the neural network model is designed to predict all the unresolved small-scale processes ${\hat{v}}_{k},{\hat{T}}_{k}$ without pre-knowledge of the original dynamical model ([Disp-formula FD6a-entropy-26-00522]). And together with the neural network output, Equation (6b) can be used to update the solution of the large-scale *U* with different noise levels $\sigma_{U}$.

#### 2.2.1. Explicit Solutions to the Topographic Model with Damping and Stochastic Forcing

The conditional Gaussian structure in the spectral flow and tracer model (6) enables to derive closed analytic formulas for discovering the typical properties in the topographic flow field and passive tracer solutions. It shows that the statistics in the velocity and tracer modes ${\hat{v}}_{k},{\hat{T}}_{k}$ in (Equation 5) can be determined by the statistics in *U*. These analytic formulas help to provide an improved understanding of the rich physical phenomena observed in both the flow and tracer fields, under which the machine learning models can be constructed. Below, we list the main conclusions, where the detailed derivations can be found in Appendix A.

First, we can find the long-time steady-state solution when the initial state becomes irrelevant. Following the decoupled dynamics from the diagonal coefficients of each wavenumber ([Disp-formula FD6a-entropy-26-00522]), the conditional trajectory solution of the steady-state small-scale mode (that is, for t≫1 large enough to ‘forget’ its initial information) can be written as
(7)$${\hat{v}}_{k}\left(t;U\left(\cdot \right)\right)=-\int_{0}^{t}e^{-\gamma_{v,k}\left(t-s\right)-ikU\left[s,t\right]}\left[{\hat{h}}_{k}U\left(s\right)ds+\sigma_{v,k}dW_{k}\left(s\right)\right],$$
with the coefficients $\gamma_{v,k}=d_{k}-ik^{-1}\beta$ and $U\left[s,t\right]\equiv \int_{s}^{t}U\left(\tau \right)d\tau$ depending on one realization of the zonal mean solution of $U\left(s\right)$ during the entire time period 0<s≤t. Similarly with the above formula for the shear flow solution ${\hat{v}}_{k}$, the corresponding solution for the passive tracer Equation ([Disp-formula FD6c-entropy-26-00522]) can be solved based on the advection flow
(8)$${\hat{T}}_{k}\left(t;U\left(\cdot \right)\right)=\frac{\alpha }{\gamma_{R,k}}\int_{0}^{t}e^{-\gamma_{T,k}\left(t-s\right)-ikU\left[s,t\right]}\left[e^{\gamma_{R,k}\left(t-s\right)}-1\right]\left[{\hat{h}}_{k}U\left(s\right)ds+\sigma_{v,k}dW_{k}\left(s\right)\right],$$
with the effective damping and dispersion relation $\gamma_{R,k}=\gamma_{T,k}-\gamma_{v,k}$. Note that the above state solutions at time *t* are dependent on the entire history of the large-scale mean flow *U*. The above explicit Formulas (Equation 7) and (Equation 8) for the trajectory solutions of flow and tracer modes imply that we can recover the small-scale flow and tracer trajectories based on the information from the large-scale mean flow information. This provides an instructive guideline for the solution structures with the small- and large-scale interaction mechanisms.

#### 2.2.2. Statistical Solutions for the Mean and Variance

Next, we compute the steady-state statistical solutions of the flow and tracer modes based on the conditional trajectory Formulas (Equation 7) and (Equation 8). We assume that the large-scale mean flow *U* reaches a statistical steady state with dominant leading order moments. Therefore, the mean and variance of the small-scale modes can be written in terms of the equilibrium mean $\bar{U}$, variance $r_{U}$, and the autocorrelation function $R_{U}$ of the large-scale flow.

Using the leading order expansion of the moments, the mean state for the shear flow modes can be computed by
(9)$$\begin{array}{cc} {\bar{v}}_{k}\equiv \langle {\hat{v}}_{k}\rangle = & -{\hat{h}}_{k}\int_{0}^{t}e^{-\gamma_{v,k}\left(t-s\right)}\langle e^{-ikU\left[s,t\right]}U\left(s\right)\rangle ds\\ = & -{\hat{h}}_{k}\int_{0}^{t}e^{i\left(k^{-1}\beta -k\bar{U}\right)s}e^{-d_{k}s-k^{2}r_{U}J_{U}\left(s\right)}\left(\bar{U}-ikr_{U}I_{U}\left(s\right)\right)ds. \end{array}$$

Above, we denote $\bar{U}=\langle U\rangle$, $r_{U}=\langle {\left(U-\bar{U}\right)}^{2}\rangle$, and the autocorrelation function $R_{U}\left(\tau \right)=r_{U}^{-1}\langle U\left(\tau \right)U\left(0\right)\rangle$ (with $\langle \cdot \rangle$ represents the statistical expectation at equilibrium). The time-dependent operators $I_{U},J_{U}$ are computed from the integration of the autocorrelation function $R_{U}$
$$I_{U}\left(t\right)=\int_{0}^{t}R_{U}\left(\tau \right)d\tau ,\;J_{U}\left(t\right)=\int_{0}^{t}\left(t-\tau \right)R_{U}\left(\tau \right)d\tau .$$

Only the first two moments of the stochastic process *U* are used in the above computation of the statistical expectation $\langle e^{-ikU\left[s,t\right]}U\left(s\right)\rangle$. Accordingly, the tracer mean state can be also derived based on the statistics of the mean flow *U* in a similar fashion using the trajectory solution
(10)$$\begin{array}{cc} {\bar{T}}_{k}\equiv \langle {\hat{T}}_{k}\rangle = & -\frac{\alpha {\hat{h}}_{k}}{\gamma_{R,k}}\int_{0}^{t}e^{-\gamma_{T,k}\left(t-s\right)}\left[e^{\gamma_{R,k}\left(t-s\right)}-1\right]\langle U\left(s\right)e^{-ikU\left[s,t\right]}\rangle ds\\ = & -\frac{\alpha {\hat{h}}_{k}}{\gamma_{R,k}}\int_{0}^{t}e^{-\gamma_{T,k}s-k^{2}r_{U}J_{U}\left(s\right)}e^{-ik\bar{U}s}\left(e^{\gamma_{R,k}s}-1\right)\left(\bar{U}-ikr_{U}I_{U}\left(s\right)\right)ds, \end{array}$$
with the new dispersion relation $\gamma_{R,k}=\gamma_{T,k}-\gamma_{v,k}$. Thus, we see that the tracer and flow means ${\bar{v}}_{k}$ and ${\bar{T}}_{k}$ are closely linked. The second-order moments of the flow and tracer modes can be computed by multiplying the corresponding states on both sides of (Equation 7) and (Equation 8) depending on the statistics and time correlations of the zonal mean flow statistics of *U*. Then, the flow velocity variance $r_{v_{k}}$ for each mode becomes
(11)rvk≡|v^k|2=12dkσv,k2+2h^k2Re∫0∞e−dk+ik−1βτU0Uτe−ikU0,τdτ.

The tracer variance $r_{T_{k}}=\langle |{\hat{T}}_{k}{|}^{2}\rangle$ can also be found similarly. The above expression for the variance is linked to the triad correlation of the large-scale steady-state flow $\langle U\left(0\right)U\left(\tau \right)e^{-ikU\left[0,\tau \right]}\rangle$. We can compute the expectation in terms of the statistics in the mean flow, that is, $\bar{U},r_{U}$ and $R_{U}$. The explicit expression can be found in Appendix A.

From the above explicit formulas for the flow and tracer statistics, we observe the already complicated structures in the leading statistics from coupling between large and small scales as well as the flow and tracer interaction. In particular, in order to resolve the mean and variance of the small-scale flow and tracer modes, it shows that detailed higher-order moments are required from the large-scale mean state *U*. This often demands huge amounts of data and expensive computational cost to achieve a desirable accuracy. On the other hand, the above mean and variance Formulas (Equation 9)–(Equation 11) with the conditional Gaussian structures imply that the essential leading-order statistical information among all small-scale processes can be recovered from the statistical measurements in the large-scale mean flow. The informed statistical solutions can offer crucial guidance for the construction of combined physics and data-driven models in the following section. Therefore, machine learning strategies will be designed to find the unresolved small-scale dynamics directly from data, while explicit physics equations will be used for capturing extreme features in the large-scale mean state. In particular, the trained machine learning model can greatly reduce the computational cost of directly running the expensive full model, and provide an efficient alternative way to predict the small-scale processes without the further requirement of intense data.

### 2.3. Different Statistical Regimes of Flow and Tracer Fields in the Two-Mode Model

Before the construction of data-driven models to learn the turbulent dynamics, we first illustrate the typical dynamical and statistical structures found in the flow and tracer solutions using direct numerical simulations. The above analytical Formulas (Equation 7) and (Equation 8) show that the flow and tracer models reach various dynamical regimes relying on mean flow statistics in *U*, while from the mean flow Equation (6b), the solution of the zonal mean flow *U* in turn is determined by the combined feedback from the small-scale fluctuation modes.

Here, we choose a prototype *two-mode topographic model* under the simplest setting using only two Fourier modes, k,2k. Accordingly, a two-mode topography can be adopted as a combination of the two scales
$$h=H_{1}\left(\cos kx+\sin kx\right)+H_{2}\left(\cos2kx+\sin2kx\right)$$

The two-mode system includes five coupling modes $\left(U,{\hat{v}}_{\pm 1},{\hat{v}}_{\pm 2}\right)$ for the flow equations and $\left({\hat{T}}_{\pm 1},{\hat{T}}_{\pm 2}\right)$ for the passive tracer state. This simple two-mode formulation keeps the central interaction mechanism between the mean flow and fluctuation modes, which is still able to create a wide range of remarkably different statistical regimes representing different kinds of extreme events. Therefore, we use this model as a basic test model to display the various distinctive statistical regimes, and then as a standard testbed for the design of neural network architectures in the following sections.

To illustrate different model statistics, our strategy is to modify the major driving effect from the topographic stress $H=H_{1}=2H_{2}$. The other model parameters are fixed as β=2, $d_{U}=d_{k}=0.0125$, $\sigma_{U}=\sigma_{k}=\frac{1}{2\sqrt{2}}$ for the flow equations, and $d_{T}=0.1$, κ=0.001 for the tracer equation. These parameters are picked according to [34,42] to simulate realistic climate scenarios. The typical trajectory solutions from direct numerical simulations are shown in Figure 1 with different topographic forcing strengths *H*. We observe the distinct dynamical structures under this simple two-mode model setup. With a weak topographic stress H=1, the mean flow displays a slow varying time scale with intermittent extreme values on the negative side. Strong extreme events are triggered in both the small-scale flow and tracer states as the large-scale *U* reaches regimes of positive values. In contrast with a strong topographic stress H=10, the mean flow *U* develops a fast oscillating time scale on top of the slow transiting packages of extreme events. The time scale difference appears more obvious between the flow and tracer fields. The tracer modes show a much slower time scale in comparison with the flow modes, and strong skewness with multiscale structures.

For a detailed comparison of the solution statistics, the probability density functions (PDFs) and autocorrelation functions (ACFs) are shown in Figure 2 for the two topographical regimes. In the strongly forced regime H=10, the mean flow *U* displays near-Gaussian statistics, while both the flow *v* and tracer *T* modes generate highly skewed PDFs. The fat-tailed or skewed PDFs are generated due to the uncertainty in the mean flow field *U* interacting with the small-scale conditional Gaussian processes ${\hat{v}}_{k}$ or ${\hat{T}}_{k}$. In the autocorrelation functions, strong scale separation is also found with a fast mixing oscillating process in the flow states for both *U* and *v*, while the tracer modes for *T* have a much more slowly decaying mixing process. This can be also observed in the time series in Figure 1. In contrast in the weak topography case H=1, the mean flow *U* develops fat-tailed non-Gaussian statistics in the PDF. The small-scale flow and tracer modes also display fat tails consistent with the time series. In the autocorrelation functions, the mean flow *U* has a much slower mixing rate in comparison with the fast-mixing modes in both the flow and tracer. The strong scale separation makes it very difficult to learn the complete model dynamics purely from data. It requires the accurate modeling of all scales at the same time during the training of the data-driven models to correctly represent the dynamics and maintain a stable scheme.

## 3. Neural Network Architecture for Correlated Dynamical Processes

In this section, we describe the general neural network architecture to learn the correlated dynamical processes in turbulent systems. Based on the model framework in Section 2, various statistical regimes can be found from the same dynamical model depending on different sets of data in the large-scale state. The main goal here is to construct an effective machine learning model to capture the complex dynamics in a uniform fashion among different statistical regimes. In the following, we propose several new structures to the basic long short-term memory (LSTM) network [43] for modeling dynamical updates with long measurement time interval Δt. A new set of loss functions based on relative entropy is also proposed to respect the turbulent nature of the extreme solution trajectories.

### 3.1. Architecture of the Neural Network Model

First, we provide a brief description of the main components in the neural network model designed to approximate a continuous dynamical system in the general form
(12)$$\frac{dy}{dt}=f\left(x,t\right),$$
where $y\in R^{n}$ is the target state to be predicted and $x\in R^{d}$ represents all the related input variables (explicit examples for the input and output data are shown next in Section 4 using the topographic model). *f* is the unknown dynamical map to be learned by the machine learning scheme from data. Adopting the general dynamical structures of the form (Equation 12), the neural network architecture is designed by (i) a residual network to approximate the increment at each time step; (ii) an LSTM chain to incorporate the correlated time sequence; and (iii) a multistage link inside each LSTM cell to capture the coupled dynamics in the model.

#### 3.1.1. Residual Network to Capture the Dynamical Model Update

In the first place, for the machine learning approximation of the dynamical increment in (Equation 12), we adopt a residual network structure for the time increment of the unresolved dynamical functional
(13)$$y_{t}^{M}=y_{t-1}^{M}+\Delta tf_{m}^{M}\left(\left\{x_{t-m},\cdots ,x_{t-1}\right\}\right).$$

Above, $f_{m}^{M}:R^{d\times m}\to R^{n}$ is the neural network approximation for the unknown dynamical model *f*. The input data consist of a correlated sequence of measurements ${\left\{x_{i}\right\}}_{i=t-m}^{t-1}$ with $x_{i}\in R^{d}$ evaluated at *m* time instants ahead of the prediction time *t*, and the new target state $y_{t}\in R^{n}$ is predicted for the next measurement time *t* from its adjacent state $y_{t-1}$. In neural network predictions, multiple previous time steps $\left\{x_{t-m},\cdots ,x_{t-1}\right\}$ are taken together as the input to include long-time correlations from the data. The updating time step $\Delta t=t_{n+1}-t_{n}$ is determined by the two adjacent measurements $x_{n}$ and $x_{n+1}$, and is usually longer than the integration step dt required for numerical stability in the direct numerical scheme of the original system (Equation 12). In addition, the input data sequence $\left\{x_{i}\right\}$ may also include measurement error and the integrated model error from the output in the previous prediction steps.

#### 3.1.2. LSTM Network to Approximate Time-Correlated Unresolved Processes

To accurately present the time series with a long memory and correlation time, we start with the basic structure of the LSTM network [43,47] for the realization of the increment updating functional $f_{m}^{M}$. The LSTM network consists of *m* LSTM cells $h_{i+1}=\mathrm{Lc}\left(x_{i},h_{i};W\right)$ with the same structure and parameters W. The cells are connected by the intermediate hidden state $h_{i}\in R^{h}$. The long-time correlation is represented by feeding the sequential data into each cell element accordingly. The full LSTM chain is linked by the *m* sequential cell structures (the connection is illustrated in Figure 3).
(14)$$h_{m}={\mathrm{Lc}}^{\left(m\right)}\left\{h_{0};x_{t-m},\cdots ,x_{t-i},\cdots ,x_{t-1}\right\}\equiv \mathrm{Lc}\left(x_{t-1}\right)\circ \cdots \mathrm{Lc}\left(x_{t-i}\right)\cdots \circ \mathrm{Lc}\left(x_{t-m}\right)\left(h_{0}\right).$$

In the above LSTM chain i=0,⋯,m−1, the input data $x_{t-m+i}$ are fed into the corresponding *i*-th LSTM cell, and $h_{i}$ is the hidden state output from the previous cell i−1 and input for the next adjacent cell *i*. For simplicity, the initial value of the hidden state is often set as zero, $h_{0}=0$ (the model dependence on the initial value appears weak with a LSTM chain of moderate length *m* from the numerical tests). The final hidden state output $h_{m}$ from the last step of the LSTM chain goes through another single layer of the fully connected network to give the model approximation of the dynamical increment for *f*
(15)$$f_{m}^{M}=\sigma \left(W^{f}h_{m}+b^{f}\right),$$
where $W^{f},b^{f}$ are the linear map and bias coefficients of the final layer, and σ is a nonlinear activation function, such as the rectified linear unit (ReLU). The detailed structure for the LSTM cell with multiple gates is listed in Section B.1.

#### 3.1.3. Modified Connections in the LSTM Cell Admitting Dynamical Structures

To adapt to the dynamical features using the LSTM net (Equation 14), we introduce additional modifications to the standard architecture. Considering that we usually have measurements at a larger time interval Δt, it is useful to introduce multiple inner update stages as the unresolved intermediate time steps to achieve higher-order accuracy in the final neural network output. In addition, using a single-stage update with the original LSTM will often lead to quick divergence in time iterations due to the inherent internal instability in the turbulent systems (such as the direct numerical tests in [27]).

We introduce a multistage structure in the one-step time update in each LSTM cell $\mathrm{Lc}\left(x,h\right)$. The idea is to fill in multiple unresolved finer time stages inside the large measurement time step Δt of the available data. It is comparable to introduce a higher-order integration scheme for the discretized dynamical Equation (Equation 12)
$$y_{t+\Delta t}-y_{t}=\int_{t}^{t+\Delta t}f\left(x\left(\tau \right)\right)d\tau \approx \sum_{l=1}^{s}b_{t,l}f_{t}^{l}.$$

Inside each LSTM cell for a one-step update of size Δt, we can generalize the cell structure to link concatenated basic layers of the inner connected LSTM units. It corresponds to building a multistage scheme in updating the present state to the next time step with a higher-order accuracy. Therefore, inside each LSTM cell *i* with input data $x_{i}$ and the input hidden state $h_{i}$, we can compute multiple output layers for each layer output j=1,⋯,s
(16)$$h_{i}^{\left(j\right)}=\mathrm{LSTM}\left(x_{i},\sum_{l=0}^{j-1}a_{jl}h_{i}^{\left(l\right)}\right),\;h_{i}^{\left(0\right)}=h_{i}.$$

Above, $a_{jl}$ are learnable model parameters for the *i*-th layer output. The intermediate stage outputs $\left\{h^{\left(0\right)},\cdots ,h^{\left(j-1\right)}\right\}$ are stacked together with the coefficients $a_{jl}$ as the input for the next stage *j*. The model parameters in the LSTM units are kept the same for different stages j=1,⋯,s since it is aimed to approximate the same dynamical functional *f* ultimately. Thus, the total size of the model parameters is not increased. The final hidden state output of this cell is computed with the combination of all the hidden layer outputs along the series of LSTM predictions:(17)$$h_{i+1}=\sum_{j=1}^{s}b_{j}h_{i}^{\left(j\right)},$$
whereas the additional coefficients $a_{jl},b_{j}$ are added to the training parameters altogether, learned directly from the data in the optimization process.

In summary, the neural network model consists of the LSTM chain (Equation 14) with *m* inputs from the previous measured states $x_{t-i},i=1,\cdots ,m$ along the time series to approximate the dynamical increment $f^{M}∼y_{t}-y_{t-1}$ at the next prediction time instant *t*. Importantly, the inner cells in the LSTM chain adopt the additional multistage scheme (Equation 17). The additional structures introduced in the model are shown to effectively improve the accuracy and robustness in both the training and prediction stages. The neural network connections are illustrated in Figure 3 for the entire LSTM chain and the inner structure of each cell.

### 3.2. Different Metrics for Calibrating the Loss Error

The last issue for the construction of the neural network is to define a proper loss function *L* measuring the error in the model prediction $y^{M}$ compared with the target $y^{t}$. In training the LSTM network (Equation 14), though only the last output is used for the prediction of the next state, the intermediate cell outputs also produce meaningful predictions for the earlier states (that is, the cell *i* with input $x_{i}$ can give a prediction for the state $y_{i+1}$ for i<m). Therefore, we measure the output sequence of the last *l* outputs (for example, the second half l=m/2) in the error metric.

The proper choice of a feasible loss function *L* also plays a crucial role to guide the optimization procedure to an efficient convergence with emphasize on both the multiscale temporal structures and the occurrence of extreme events along the time series. We compare three different choices for the cost $L\left(x,y\right)$ to calibrate the difference between the model output $x≐y^{M}$ and the truth target $y≐y^{t}$:The $L_{2}$ distance:
(18a)$$L_{2}\left(x,y\right)={∥x-y∥}_{2}^{2}=\frac{1}{M}\sum_{j=1}^{M}{\left|x_{j}-y_{j}\right|}^{2};$$The relative entropy loss:
(18b)$$L_{\mathrm{KL}}\left(x,y\right)=\frac{1}{M}\sum_{j=1}^{M}\sum_{i}{\tilde{y}}_{i}^{\left(j\right)}\log\frac{{\tilde{y}}_{i}^{\left(j\right)}}{{\tilde{x}}_{i}^{\left(j\right)}};$$Mixed loss by an $L_{2}$-relaxation with the relative entropy loss:
(18c)$$L_{\mathrm{mix}}\left(x,y\right)=L_{\mathrm{KL}}\left(x,y\right)+\alpha L_{2}\left(x,y\right).$$

The $L_{2}$ distance ([Disp-formula FD18a-entropy-26-00522]) measuring the mean square error is the most common choice of loss function, comparing the pointwise measurements of errors. In the case with data from turbulent models, small fluctuations in the solutions may end up with an unnecessarily big contribution in the $L_{2}$ loss. The pointwise measurement thus may add too much emphasis on the accumulated errors from small turbulent fluctuations. On the other hand, we aim to capture the dominant emerging features such as the extreme events. Thus, the relative entropy loss ([Disp-formula FD18b-entropy-26-00522]) enjoys the advantage of focusing on the main coherent features of the solution invariant under small shifts in the extreme value locations. The input data in ([Disp-formula FD18b-entropy-26-00522]) are also rescaled and measured separately from the partition functions ${\tilde{x}}_{i}^{\pm }=\frac{\exp\left(x_{i}/T_{\pm }\right)}{\sum_{i}\exp\left(x_{i}/T_{\pm }\right)}$, with $T_{+}>0,T_{-}<0$ weighting the importance of the positive and negative extreme values as suggested in [31].

The mixed cost function in (18c) is shown to enjoy the advantages of the two forms of cost functions ([Disp-formula FD18a-entropy-26-00522]) and ([Disp-formula FD18b-entropy-26-00522]). The combination accounts for both the small-scale fluctuation and the dominant extreme events in the turbulent solution. The $L_{2}$-relaxed form is useful to fit the various small-scale structures in the solution, while the relative entropy emphasizes the extreme values in the solutions. In practice, the parameter α can take a relatively small value just as a penalty term (we pick α=0.1 in all the following numerical experiments as an empirical choice from different tests; it is worthwhile to carry out a systematic study for the choice of this parameter). The modified model with the above multistage inner connection in the LSTM cell is show to allow a larger learning rate in the stochastic gradient descent process, so it enjoys faster convergence and stability compared with the original models. The method is shown to be fairly robust to the choice of hyperparameters.

## 4. Learning Multiscale Dynamics Informed of the Physical Model

Now, we apply the general machine learning model discussed in Section 3 for the prediction of multiscale dynamical systems with extreme events. The clean structures in the unambiguous model (6) provide a desirable testbed with rich statistical features. According to the conditional Gaussian properties shown in Section 2, the neural network is designed for multiscale processes with unstable interactions. The optimized network is shown to have robust performance among different statistical regimes (explicit examples will be shown next in Section 5). The strategy is also generalizable to a wider class of complex systems with interacting scales.

### 4.1. General Model Setup and the Neural Network Model

For the topographic barotropic model (6), depending on the realization of the large-scale mean flow solution *U*, the trajectory solutions for the flow and tracer modes can be computed by integrating the exact Equations ([Disp-formula FD6a-entropy-26-00522]) and ([Disp-formula FD6c-entropy-26-00522])
(19)$$\begin{array}{cc} {\hat{v}}_{k}\left(t+\Delta t\right)= & \;{\hat{v}}_{k}\left(t\right)+{\hat{F}}_{k}^{v}\left(U\left(\cdot \right),{\hat{v}}_{k}\right)={\hat{v}}_{k}\left(t\right)+\int_{t}^{t+\Delta t}\left\{\left[i\left(k^{-1}\beta -kU\right)-d_{k}\right]{\hat{v}}_{k}-{\hat{h}}_{k}U\right\}ds,\\ {\hat{T}}_{k}\left(t+\Delta t\right)= & \;{\hat{T}}_{k}\left(t\right)+{\hat{F}}_{k}^{T}\left(U\left(\cdot \right),{\hat{v}}_{k},{\hat{T}}_{k}\right)={\hat{T}}_{k}\left(t\right)+\int_{t}^{t+\Delta t}\left[\left(-\gamma_{T,k}-ikU\right){\hat{T}}_{k}-\alpha {\hat{v}}_{k}\right]ds. \end{array}$$

Above, we use ${\hat{F}}_{k}^{v}$ and ${\hat{F}}_{k}^{T}$ to represent the unresolved dynamics in the fluctuation modes in different scales. The tracer modes ${\hat{T}}_{k}$ are passively advected by the advection flow field $\left(U,v\right)$, while the flow modes ${\hat{v}}_{k}$ also give a combined contribution to the evolution of the large-scale mean flow state *U*. With solutions for the unresolved states, we can then compute the zonal mean flow solution by directly integrating Equation (6b) forward in time
(20)$$U\left(t+\Delta t\right)=U\left(t\right)+\int_{t}^{t+\Delta t}f_{U}\left(v\right)d\tau +\sigma_{U}\sum_{j=1}^{s}\Delta W_{U,j}.$$

Above, the deterministic component on the right-hand side of the equation is summarized in the operator $f_{U}\left(v\right)=\sum_{k}h_{k}^{*}{\hat{v}}_{k}-d_{U}U$. The first term represents the topographic stress from the combined feedback from all the small-scale modes, and the second term serves as a friction from the boundary. The last white noise forcing introduces additional uncertainty to the system with different noise amplitudes $\sigma_{U}$. In addition, the time integration about $f_{U}\left(v\right)$ also includes model approximation error since we only have the model output from the neural network at two adjacent time measurements at *t* and t+Δt, and the data observation time Δt is often larger than the desired integration time step dt.

We then apply the general neural network framework (Equation 13) to the specific set of Equation (19) for both the flow and tracer modes. In practice, the input data can be chosen to include all state variables, including both large- and small-scale flow and tracer modes $x=\left\{U,{\hat{v}}_{k},{\hat{T}}_{k}\right\}$, and the prediction targets are set as the unresolved small-scale flow and tracer modes $y=\left\{{\hat{v}}_{k},{\hat{T}}_{k}\right\}$ at the next prediction time among all the wavenumbers *k*. The length of the LSTM net *m* can be determined according to the decorrelation time of the state so that $m\Delta t∼T_{\mathrm{decor}}$, where $T_{\mathrm{decor}}=\int R\left(s\right)ds$ is the integrated autocorrelation function characterizing the mixing time scale.

#### 4.1.1. Decoupled Neural Networks for Multiscale Dynamics

As we have shown in the model analysis in Section 2, one of the main challenges in learning the turbulent dynamics is to resolve the strong scale separation between the coupled processes. The smaller scale mode with a larger wavenumber usually exhibits faster mixing time and smaller variance (see from the explicit formulas and numerical examples in Section 2.3). For efficient modeling, we can decompose the high-dimensional systems according to the different scales, and propose a neural network model focusing on one specific scale structure so that the multiscale structure is better represented. The resulting neural networks also become easier to train since we decompose the large system into several smaller subsystems, requiring fewer model parameters.

Exploiting the conditional Gaussian structure, small-scale modes ${\hat{v}}_{k},{\hat{T}}_{k}$ in (19) become decoupled conditional on the large-scale mean state *U*, while the mean state *U* is updated by the physical model (20) from the combined feedback from different scale modes. In particular, for the two-mode model, it is natural to propose two separate neural networks for the wavenumbers for capturing different scale dynamics which are coupled finally through the large-scale dynamics of *U*. The general framework for learning the dynamical structure of the two-mode topographic barotropic system (6) can be modeled together using the multiscale neural network models $\left\{f_{k}^{M}\right\}$ and the explicit physical dynamics for the observed large-scale state *U*
(21)$$\begin{array}{cc} y_{n+1}\left(k\right) & =y_{n}\left(k\right)+f_{k}^{M}\left(\left\{x_{n-m:n}\left(k\right)\right\};W\right); \end{array}$$
(22)$$\begin{array}{cc} U_{n+1} & =U_{n}+\Delta tF_{U}\left(\left\{y_{n,n+1}\right\}\right)\;+\sigma_{U}\sqrt{\Delta t}\xi_{n}. \end{array}$$

Above, the neural network model $f_{k}^{M}$ in (21) is used to learn the detailed single-scale dynamical updates in each small-scale mode. The large-scale dynamics is integrated explicitly using the physical dynamics from the output data of the neural network. Thus, the true large-scale physical dynamics can be introduced directly to the training model. $F_{U}\left(y\right)$ gives the approximation for the deterministic integration, while additional uncertainty is introduced from the white noise forcing ξ. For computing the explicit integration, we use the mid-point rule combining the input and output data
(23)$$\Delta tF_{U}\left(y_{n,n+1}\right)=\int_{t}^{t+\Delta t}f_{U}\left(v\right)d\tau =\frac{\Delta t}{2}\left[f_{U}\left(v_{n}\right)+f_{U}\left(v_{n+1}\right)\right],$$
where $v_{n}$ is the model input and $v_{n+1}$ is the model prediction at the next time step for the numerical integration of the term $f_{U}\left(v\right)$ in (20). In this way, the predicted state *U* will incorporate information from the neural networks and the large-scale physical process.

As a remark, above, we apply the neural network to the example of the simplest two-mode model so that we are able to carry out a detailed investigation for the various features in the neural network model using the explicit model solutions. However, the idea for modeling multiscale structures can be easily generalized to higher-dimensional systems with multiple interacting scales. For example, a high-dimensional system can be decomposed into block-diagonal subsystems using the conditional Gaussian framework [24,42]. Then, the neural network model can be applied to each block approximating a series of subsystems focusing on different scales.

#### 4.1.2. Multistep Training Loss Including Time-Dependent Data

Above, the model (21) and (22) gives a one-step prediction with a time step size Δt for the long time series. In practice, we would like to use the trained network model for long-time prediction even beyond the decorrelation time of the system. Then, there comes the problem of numerical stability and robustness from the accumulation of model approximation errors amplified through the repeated updating steps with strong internal model instability. To address this issue, a multistep model output is used in optimizing the loss function during the training process.

In training for the turbulent model with high degrees of instability, the loss function is expected to guide the neural network to gain the skill to detect the occurrence of bursting extreme events as well as the complex structures in the dynamical model. Therefore, instead of simply training the model from a one-step output, we iterate the system (21) and (22) forward in time up to *n* steps using the model output as the initial data for the next iteration. The general form of the loss function can be designed by the total loss along the *N* updates with the proper cost *L* in (18)
(24)$$L_{n}=\sum_{i=1}^{n}w_{i}L\left(F_{M}^{\left(i\right)}\left(x;W\right),y^{\left(i\right)}\right).$$

Above, the LSTM model (21) is trained online, combining the output from the physics model (22) during the time iterations. $F_{M}^{\left(i\right)}$ is the push forward operator from recursively running the model up to nΔt, and $y^{\left(i\right)}$ is the target state to be compared. The weights $w_{i}$ offer a balanced calibration of the model output series. A convenient choice of the weights is to use the autocorrelation function $w_{i}=\left|R_{y}\left(i\Delta t\right)\right|$ for the corresponding state in the measurement. This provides a balanced quantification for the prediction error, where the prediction for longer future time is tolerated with the smaller weight $w_{i}$. Note that the ‘multistep’ training here involves time updating with *n* iterations, which is different from the ‘multistage’ neural network architecture in Section 3.1 inside one time update Δt.

To exploit the conditional Gaussian structure of the two-mode model with different statistics, we divide the data for training and prediction according to the large-scale mean flow *U*. Based on the explicit solutions, data in a moderate regime (with a relatively small white noise forcing $\sigma_{U}$) are used to train the model parameter under the loss function (24). The trained model is then applied to predict the solutions among different statistical regimes from near-Gaussian to highly skewed PDFs according to different noise levels of $\sigma_{U}$ (see the results in Section 5.3 and Appendix B). The model is shown to be robust to model errors from multistep iterations and noise from the above multistep training strategy.

### 4.2. Metrics to Measure the Accuracy in Training and Prediction

Finally, the proper metrics should be proposed to calibrate the accuracy in the model training results and predictions. In the training stage with a batch of *M* samples, the neural network is iterated *N* steps for a sequence of model outputs at time $t_{n}=n\Delta t,n=1,\cdots ,N$. One direct way to compute the training error is through the $L_{2}$ distance among all the *M* training samples and in the *N*-step outputs. The *batch averaged relative mean square error* (BMSE) can be defined as
(25)$$\mathrm{BMSE}=\frac{1}{M}\sum_{i=1}^{M}\frac{\sum_{n=1}^{N}{∥y_{m,i}^{\left(n\right)}-y_{t,i}^{\left(n\right)}∥}^{2}}{\sum_{n=1}^{N}{∥y_{t,i}^{\left(n\right)}∥}^{2}},$$
where $y_{m,i}^{\left(n\right)}$ is the model output in the *i*-th sample and at the *n*-th step model output, and $y_{t,i}^{\left(n\right)}$ is the corresponding target state at time step $t_{n}=n\Delta t$. The relative error is normalized by the $L_{2}$-norm for the true state ${∥y_{t,i}^{\left(n\right)}∥}^{2}$ where all the samples are summed with index *i*, and thus the BMSE error measures the training accuracy according to the total variability among the batch samples.

In the prediction stage, we need to track the development of errors in time when the solution is achieved by recursively iterating the optimized model using the outputs. First, for the trajectory prediction for a long time series, we can calibrate the trajectory error at each single prediction time *t* by comparing the error with the equilibrium variance of the state. The *normalized mean square error* (NMSE) for prediction accuracy can be defined by the averaged error of *M* test samples
(26)$$\mathrm{NMSE}\left(n\right)=\frac{1}{M}\sum_{i=1}^{M}\sigma_{y}^{-2}{∥F_{M}^{\left(n\right)}\left(x_{0,i}\right)-y_{t,i}^{\left(n\right)}∥}^{2},$$
where $\sigma_{y}^{2}=\langle {\left|y_{t}\right|}^{2}\rangle$ is the variance of the predicted state at equilibrium. $F_{M}$ is the optimized model operator, and the prediction at the time t=nΔt from the initial state $x_{0}$ is computed by iterating the model *n* times. Next, it is also useful to measure the statistical accuracy in mean and variance from the statistical model with ensemble solutions. Therefore, we can define the *statistical mean error* (SME) and *statistical variance error* (SVE) as
(27)$$\mathrm{SME}=\frac{{\left|\langle y_{m}\rangle -\langle y_{t}\rangle \right|}^{2}}{\langle {\left|y_{t}^{\prime }\right|}^{2}\rangle },\;\mathrm{SVE}=\frac{\left|\langle {\left|y_{m}^{\prime }\right|}^{2}\rangle -\langle {\left|y_{t}^{\prime }\right|}^{2}\rangle \right|}{\langle {\left|y_{t}^{\prime }\right|}^{2}\rangle }.$$

Above, we use $\langle f\rangle =\frac{1}{M}\sum_{i=1}^{M}f_{i}$ to represent the statistical expectation computed by averaging among the samples. $\langle y_{m}\rangle ,\langle y_{t}\rangle$ are the ensemble means for the model prediction and target data statistics; $\langle {\left|y_{m}^{\prime }\right|}^{2}\rangle ,\langle {\left|y_{t}^{\prime }\right|}^{2}\rangle$ are the variances about the fluctuation states $y^{\prime }=y-\langle y\rangle$. Instead of the pointwise measurement of errors in (26), the statistical errors in (27) calibrate the model’s skill in capturing the representative statistical moments in an ensemble solution.

## 5. Predicting Extreme Events and Related Statistics Using the Neural Network Model

In this section, we study the learning and prediction skill in the developed neural network model guided by the tractable model framework of the coupled topographic flow and tracer system. As analyzed in the explicit formulas in Section 2, the simple system is able to generate a variety of regimes with distinctive statistics and various extreme events. The new model architectures described in Section 3 are applied to capture the essential multiscale dynamics in both the turbulent flow and passive tracer fields at the same time, using a single set of data. In particular, we focus on two representative statistical regimes with strong H=10 and weak H=1 topography (as illustrated in Figure 1).

The data are collected through a long-time simulation of the original topographic model (6). The entire dataset is divided into two sections with short time trajectories for training and a new set of long time series to check the prediction accuracy:

In the *training stage*, the time trajectory is segmented into batches of short sequences for training the neural network model parameters. Usually, the model is only updated a small number of steps of length nΔt (say, n=10 in the standard test case) for efficient training.In the *prediction stage*, the optimized model is used for prediction along a long time sequence. The prediction model is iterated recurrently using the previous outputs up to a long-time NΔt (say, N=50,000 iterations).

In the training stage, a huge computational overload will be generated if we measure the errors in the model outputs for too many iteration steps because we need to back-propagate all the way to the first step in computing the gradient of the final loss function (24). Therefore, a smaller number of model update *n* is preferred for saving memory and efficiency. In the prediction stage, on the other hand, we need to keep iterating the model output for the next forecast step to a long-time prediction time. Thus model approximation errors will accumulate in time, and this requires stability in the constructed model, especially for the highly turbulent regime with strong inherent internal instability.

In the benchmark model, we fix the standard model hyperparameters as the input LSTM chain length m=100, the hidden state in the LSTM cell h=50, measurement time step in the data Δt=0.1 (which is 10 times the numerical integration step dt=0.01 in the direct numerical scheme). In the multistep training loss in (24), the training model is iterated forward n=10 times. This accounts for a time length T=nΔt=1 which is still much shorter than the decorrelation time of the slow modes (see the autocorrelation functions in Figure 2). The detailed neural network model configuration and parameters are listed in Section B.2.

### 5.1. Training and Prediction with Different Loss Functions

First, we compare the training performance under different metrics of the loss functions to measure the training error. During the training procedure, we generate a long time series with 10,000 measurements in time, and divide the dataset into smaller batches with batch size 100 to train each batch with 100 iterations. Then, the model is trained repeatedly over the same data for 100 epochs. The learning rate starts with lr=0.005, and is decayed by 0.5 of its previous value twice, at epoch numbers 50 and 80.

Here, the models with same architecture and hyperparameters are trained under the same set of data but optimized using the three different types of loss functions ([Disp-formula FD18a-entropy-26-00522])–(18c) as the optimization target. The $L_{2}$ mean square error ([Disp-formula FD18a-entropy-26-00522]) is the most common choice for the error metric, while for training turbulent signals with intermittency and extreme events, it is found that the introduction of the KL divergence ([Disp-formula FD18b-entropy-26-00522]) is effective for capturing the dominant extreme values [31]. In the mixed metric (18c) for loss, the KL divergence is still taking the dominant role with the $L_{2}$ error creating an additional balancing effect with a small weight α=0.1.

The training errors under different loss functions are compared in the upper panel of Figure 4. We compute the batch averaged mean square errors (25) for the flow and tracer states during training iterations for the two test regimes with strong and weak topography H=10 and H=1. The BMSE represents the averaged relative error in the training batch, so it serves as an index for training accuracy in each epoch. In general, all the three metrics are effective to improve the model accuracy through training iterations. The mixed metric gains the highest overall training accuracy among the flow and tracer states, and decays to a smaller error value in a faster rate than that of the other two metrics. The faster convergence implies that the mixed metric loss function is easier and more efficient to train to reach a higher accuracy with much fewer required training epochs.

Next, to check the prediction skill and stability of the trained neural network models optimized under the three different loss functions, we compare the errors (26) under a new dataset for the prediction of a relatively long time series of L=500 steps (that is, up to time T=LΔt=50 far beyond the model decorrelation time, see the autocorrelation functions in Figure 2). The model prediction in the previous time step is used recurrently for the prediction in the next time instant; thus, prediction errors will accumulate in time. One important issue is to check whether the trained neural network model can stay robust to the increasing level of errors in the input data. The lower panel of Figure 4 compares the evolution of prediction errors during the time updates. In the weak topography case H=1, the three optimized models all stay stable within finite errors during the iterative updates. Still, the optimal model trained with the mixed loss function gives the most accurate long-time accuracy with the smallest error in both flow and tracer states during the entire time. The models with the other two loss metrics generate much larger prediction errors for the longer prediction range. In the strong topography case H=10, the stronger coupling between the large and small scales increases the instability in the system. The models under the MSE and KLD loss perform well only in the starting time but eventually become unstable as the errors diverge to infinity. In contrast, the model under the mixed loss function stays stable with high accuracy during the entire prediction process. This further confirms the crucial role in selecting the proper loss function for the robust performance of turbulent systems with instability.

### 5.2. Maintaining Model Stability by Measuring Multiple Forward Steps

Another important issue we would like to check is the number of forward steps *n* used in training the loss function (24)
$$L_{n}=\sum_{i=1}^{n}w_{i}L\left(F_{i\Delta t}\left(x\right),y_{i\Delta t}\right),$$
where the model *i*-th step output $F_{i\Delta t}\left(x\right)$ is compared with the target data $y_{i\Delta t}$ under the most effective mixed loss metric *L*, and $w_{i}=R\left(i\Delta t\right)$ is the weighting factor from the autocorrelation function. The most straightforward way is to measure the error in a single-step LSTM net output n=1. However, as we have discussed in the previous section, this may lead to severe instability with exploding errors for long trajectory iterations of the model.

To address this problem, we adopt the multistep training strategy using multiple model outputs to be measured in the loss function $L\left(x,y\right)$ with n>1. In order to capture the dynamical variability in a longer time range, we iterate the training model forward in time with moderate steps n=10. The errors will accumulate in time, and the loss function at different forward times is compensated for by the decaying autocorrelation function $R\left(t\right)$. As a cost, more memory will be consumed during the back-propagating of the entire network. However, as we will show next, the cost can be controlled by using an even smaller number of steps (such as n=5) to reach the desirable training performance.

#### 5.2.1. Training and Prediction Errors with Different Forward Time Steps

Again, we first compare the training performance in the models measuring different forward steps in the training metric. We focus on the improvement by using a multistep forward model n=5,10 in contrast with a single-step training n=1. Figure 4 displays the training loss and accuracy during the training iterations. The value of the loss function decays faster to the final optimized level in the multistep training cases. And a larger forward step n=10 improves the overall accuracy compared with a smaller step n=5. For the training accuracy, the single-step model n=1 suffers a barrier to reach high accuracy compared with the other two models with multiple steps n=5 and n=10. The two models with different forward steps n=5,10 can reach comparable amplitudes of final training error, while the multistep model n=10 gains a higher accuracy and stability for long trajectory prediction.

Then, we compare the prediction skill in the three trained models on a new set of data for model evaluation. As previously, we run an ensemble prediction and recursively iterate the model outputs for the next step prediction up to the final prediction time T=50 (with 500 iterations). The lower panel of Figure 5 compares the normalized MSEs using trained models with one-step n=1 and multistep n=5,10 training. The multistep trained models gain very high accuracy in both the flow and tracer modes. The errors stay small during the entire prediction process; thus, the model approximation errors from the previous steps will not damage the accuracy in the future forecasts, implying the robustness of the trained model. On the other hand, the one-step trained model becomes insufficient to maintain the accuracy as the forecast step increases. The predicted solution stays accurate for the starting period of time, then diverges as the errors accumulate in time. Notice the logarithmic scale in the *y* coordinate for a much larger error in the n=1 case.

#### 5.2.2. Trajectory Prediction Including Multiple Time Scales

In addition, to provide a better illustration for the predicted solutions from models with different training steps, we plot one realization of the trajectories using the trained models with single and multistep training. Figure 6 shows one typical solution trajectory compared with the truth from the direct model simulation. The variance of the sample errors is plotted by the shaded area around the solution line to characterize the uncertainty in the ensemble prediction. Consistent with the previous observation from the errors, the multistep training model has good agreement with the truth in the prediction through the large number of model iterations. The multistep trained model also maintains the overall accuracy with very small variance in errors.

In contrast, the single-step trained model can only stay accurate for a short time from the initial state, while the model approximation errors quickly drive the model prediction away from the target trajectory as errors grow in time during the iterations. Large deviation from the truth is developed in time gradually, and the samples give increasingly large variance among the errors from different trajectory predictions. In particular, in the tracer time series, the solutions display two contrasting time scales with a fast oscillating small scale on top of the long-time slowly varying profile. The one-step trained model fails to track the rapid variability in the solution, and strong instability in the system leads to large error variances. The multistep training model, on the other hand, captures both time scales with very small error variance for uniformly high accuracy and stability among all the samples during the entire simulation time.

### 5.3. Long-Time Prediction in Different Dynamical Regimes

In the final test, we check the model prediction for the long-time trajectory and statistical solutions of the advection flow states and the passive tracers. This is first used to confirm the long-time stability of the neural network scheme with errors accumulated in time through the time iterations; next, the same trained model is used to predict the key statistical features with different noise-forcing levels to show the scope of skills in the model. To evaluate the prediction accuracy in the neural network models, we adopt two approaches by examining both the deterministic trajectory solution and the statistical solution:

*Trajectory prediction*: the neural network is used for the pathwise prediction for one trajectory solution from a particular initial state with uncertainties from input value and white noise forcing. Solution trajectories are solved very efficiently with *N* = 50,000 iterations (with time step size Δt=0.1 to the final simulation time T=5000).*Statistical prediction*: the neural network is used to recover from data key statistical features in leading-order statistics generated by different white noise forcing amplitudes. Instead of focusing on the pathwise solutions, it is often more useful with practical importance to learn the representative statistical structures directly.

In confirming the universality of the trained model, the neural network model is trained using a limited dataset from a single white noise-forcing regime $\sigma_{U}{\dot{W}}_{0}$ of moderate amplitude $\sigma_{U}=10\sigma_{0}=\frac{1}{2\sqrt{2}}$ in the large-scale flow equation. The small-scale flow and tracer dynamics are learned based on this dataset, then the trained model is applied for the various dynamical regimes by changing the noise forcing amplitude $\sigma_{U}$. This can also serve as a way to confirm that the true dynamical structure is indeed learned from the model rather than being due to purely the overfitting of the data.

#### 5.3.1. Trajectory Prediction with the Trained Model in Different Dynamical
Regimes

In the trajectory prediction through the neural network model, we first check the detailed prediction skill for the multscale flow and tracer structures and the occurrence of extreme events. Even though high instability exists, preventing the long-time predictability, we can exploit the conditional Gaussian structure of the model and achieve accurate pathwise prediction. The conditional solution of the small-scale solutions ${\hat{v}}_{k}\left(U\left(\cdot \right)\right)$ together with the passive tracer modes ${\hat{T}}_{k}\left(U\left(\cdot \right)\right)$ can be predicted from the trained neural network efficiently given the observed solution of the large-scale process *U*.

The long-time trajectory predictions in the weak and strong topographic stress regimes and the noise forcing $\sigma_{U}=10\sigma_{0}$ are shown in Figure 7 and Figure 8. The true model solution and statistics can be found in Figure 1 and Figure 2 in Section 2. The solutions display very strong intermittent bursts of extreme events in both small-scale flow and tracer models companied by a very fast oscillating scale. Strong time-scale separation also exists between the advection flow states $\left(U,v\right)$ and the passive tracer *T*. The neural network model is shown to accurately track such features in time and keep the high accuracy during the model iterations for the entire duration. This trajectory prediction shows explicitly the skill of the proposed neural network model to learn and recover the true model dynamics.

In addition, the same trained neural network model can be used for the prediction with different white noise-forcing levels $\sigma_{U}$ in the large-scale mean flow Equation (6b). Notice that the small-scale Equation ([Disp-formula FD6a-entropy-26-00522]) to be learned from the neural network has the same fixed dynamics, while different noise levels $\sigma_{U}$ can induce distinct statistical features in both small- and large-scale states. This guarantees the validity of the trained model among different statistical regimes once the essential dynamics is learned from data. In Section B.4, we show the prediction results with a smaller or larger effect of white noise forcing $\sigma_{U}=\sigma_{0}$ and $\sigma_{U}=20\sigma_{0}$ using the same trained neural network model to recover the unresolved small-scale solutions. It is further confirmed the universal prediction skill in the trained model among different statistical regimes, and the optimal performance is not purely through the overfitting of the training data.

#### 5.3.2. Statistical Prediction in Leading Statistics for Different Noise
Forcing

At last, we show the skill in the neural network model to recover the model statistics among various statistical regimes. By inspecting the analytical analysis results from the original model and the direct simulations of the original model, a wide variety of distinct statistics are generated under the same model framework. The same trained neural network model is then applied to predict the statistical features under these different forcing scenarios by varying the white noise-forcing amplitude $\sigma_{U}$. The conditional dynamics for the small-scale flow and tracer states stay the same for different noise levels. The question is whether the trained neural network model is capable of recovering the different statistics in a uniform way from changing the noise-forcing strength $\sigma_{U}$ for the large-scale mean state.

The model is trained based on a single set of data, being especially unaware of the strong extreme event regimes with a large amplitude from a large forcing. To measure the accuracy in the prediction of the leading statistics, we use the relative statistical error metrics (27) for measuring the accuracy in the ensemble mean and variance. In the tests, we run an ensemble prediction of M=5000 trajectories and iterate the model for N=500 steps. The statistics are computed from the model output in the last 200 steps when the steady state is reached. The neural network model enjoys the advantage to run a large ensemble very efficiently compared with the direct simulation.

Table 1 lists the statistical prediction for systems with different white noise-forcing strengths $\sigma_{U}=10\sigma_{0},20\sigma_{0}$. In general, the neural network model shows uniformly high skill in recovering the leading statistics in mean and variance among the different statistical regimes. The statistical mean error (SME) calibrates the deviation in the ensemble mean from the ensemble prediction in each of the flow and tracer modes. The statistical mean error stays in small values and keeps very high accuracy. The statistical variance (SVE) calibrates the deviation in the ensemble variance compared with the truth. This characterizes the model uncertainty in each mode, and thus is a more interesting quantity to measure. The modes become more energetic with higher uncertainty, as the white noise forcing $\sigma_{U}$ increases. The statistical errors grow with the larger value of $\sigma_{U}$, while they all stay as small values for an overall accurate statistical prediction.

## 6. Summary and Discussion

We study effective machine learning strategies to predict the various anomalous statistics and the occurrence of extreme events in complex turbulent systems using an unambiguous model framework. The model is derived from the geostrophic barotropic flow and turbulent passive tracer transport [29,34] that share many similarities with natural and laboratory observations [8,54]. The coupled system is characterized by interacting multiscale processes in time and space, leading to very complicated dynamical structures. The attractive statistical features include exact formulas for flow and tracer solutions, explicit nonlocal structures in flow and tracer modes, and the intermittent probability distributions with fat-tails and skewed PDFs. The tractable solutions of the model framework facilitate a systematic analysis of crucial multiscale properties with both mathematical theories and the development of novel numerical strategies. Detailed data-driven models are constructed to recover these statistical features from limited observed data combined with model noises.

We consider two approaches of using machine learning techniques to predict representative model solutions and statistics based on the conditional Gaussian properties: (i) a neural network to learn the unresolved small-scale dynamics and directly predict the trajectory solutions guided by the conditional Gaussian framework, and (ii) using the neural network to recover the crucial statistical moments among different regimes. New architectures are designed on the LSTM network with a residual network structure to predict the dynamical update of the unresolved small-scale states. The individual model outputs in different scales are combined in the explicit large-scale mean flow equation to inform the model with physical dynamics. Multiscale effects in large and small scale flow states as well as in the tracers are considered in the model construction and training procedures. We find the major observations in model performance using the simple two-mode topographic model:

The trained neural network model shows uniformly high skill in learning the true dynamics. The improved model architecture enables a faster convergence rate in the training stage, and more accurate and robust predictions under different forcing scenarios. A longer time updating step is permitted allowing data measured at sparse time intervals.The choice of a proper loss function for the optimization of model parameters is shown to have a crucial role to improve the accuracy and stability in the final trained neural network. A mixed loss function using the relative entropy loss together with a small $L_{2}$ loss correction is shown to effectively improve the accuracy in training for complex systems with extreme events.A multistep training process, that is, using multiple iterative model outputs in training the loss function, is useful to improve model stability against the accumulated model errors during long-time iterations. The prediction skill of the model can be improved, and training efficiency is maintained by measuring only small update steps during the training procedure.The solution trajectory can be tracked by the neural network model with high accuracy and stability in a long time series prediction for the key multiscale structures with extreme events in flow and tracer states.Different model statistics in ensemble mean and variance can be predicted with uniform accuracy among different dynamical regimes using the same neural network model trained from a single set of data.

The promising results just set the starting point for a series of interesting future research directions for the next stage. The neural network model provides the exact structure to incorporate the conditional Gaussian framework and multiscale nonlinear dynamics. A direct generalization is to use the neural network model to prediction explicit higher-order statistics as well as the non-Gaussian PDFs in the flow and tracer fields. This framework is also ready to be generalized to a wider group of complex models with a large number of interacting modes and contributions from different scales by assigning the neural networks to capture processes with different scales. This conditional independent construction of models is easy to parallelize and thus becomes especially convenient for the implementation on GPUs. The neural network framework also shows potential to be combined with the linear and nonlinear response theories for the development of statistical data assimilation and control of high-dimensional systems [55,56,57]. Thus, efficient statistical model reduction strategies [26,27] can be directly applied to learn the dynamical structure from the nonlinear interactions directly from data.

## Figures and Tables

**Figure 1 entropy-26-00522-f001:**
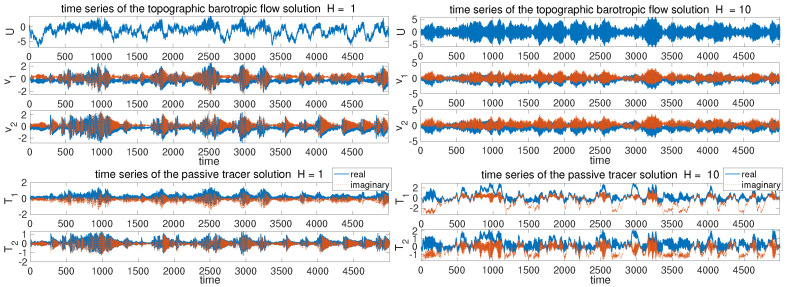
Time series of the flow and tracer trajectory solutions in the two parameter regimes with weak topographic stress H=1 (**left**) and strong topographic stress H=10 (**right**).

**Figure 2 entropy-26-00522-f002:**
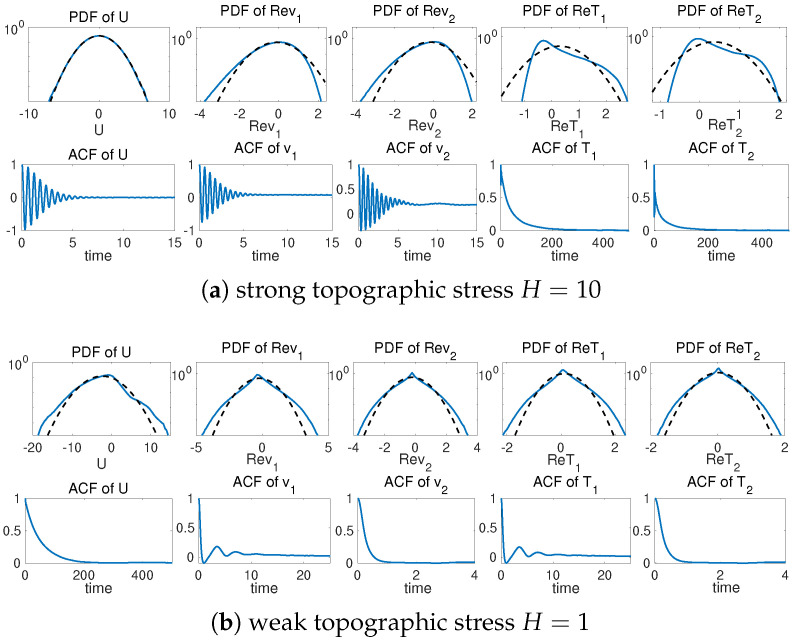
PDFs and ACFs for flow and tracer states with strong (**upper**) and weak (**lower**) topographic stress *H*. The results for large-scale state *U* and the leading small-scale modes ${\hat{v}}_{k},{\hat{T}}_{k}$ are compared. The Gaussian fits of the PDFs with the same mean and variance are plotted in dashed lines.

**Figure 3 entropy-26-00522-f003:**
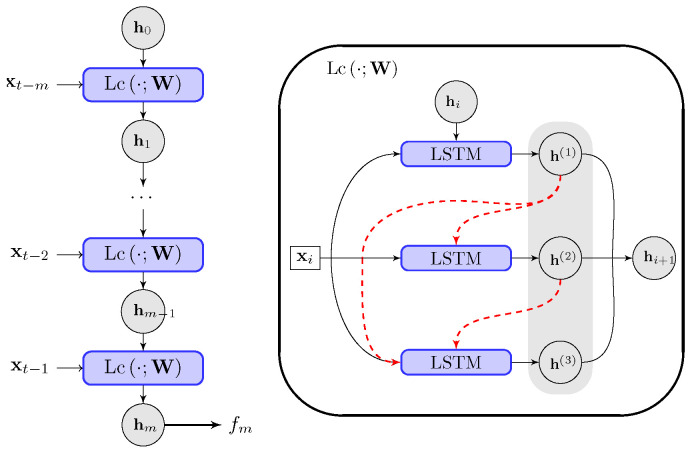
Illustration of the connection for the LSTM chain and the inner structure in each LSTM cell with three inner stages of the same structure.

**Figure 4 entropy-26-00522-f004:**
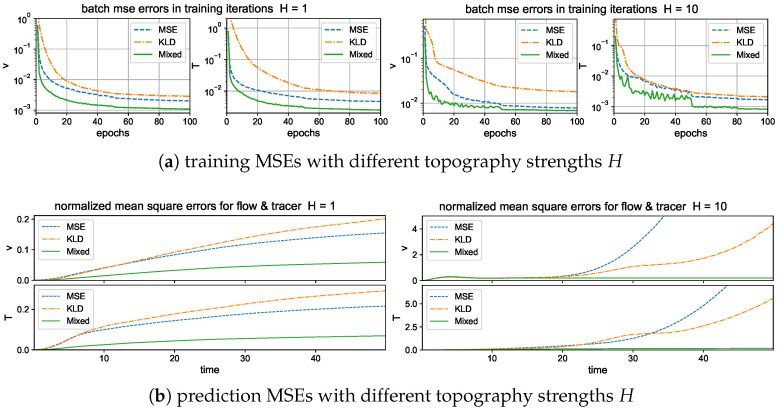
Comparison of training and prediction errors under three different loss functions *L* for optimization: $L_{2}$ loss (MSE), KL divergence (KLD), and the mixed loss combining these two. Training iterations of the batch averaged mean square errors are shown in the upper panel with logarithmic scale along the *y* coordinate. The prediction normalized mean square errors are compared in the lower panel measured at recurrent predictions of 500 time steps up to T=50.

**Figure 5 entropy-26-00522-f005:**
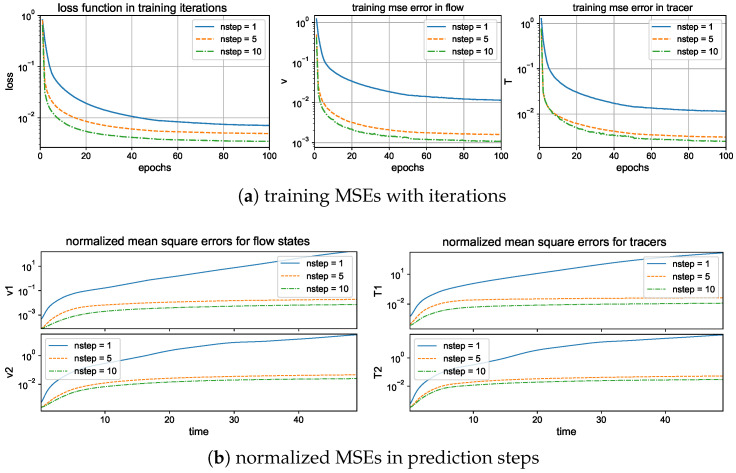
Upper: the evolutions of loss function values and the BMSEs during training iterations for models with different training forward steps n=1,5,10. Lower: normalized MSEs in model prediction with an ensemble prediction of M=500 samples up to time T=50. Performance of the trained models with different forward time steps n=1,5,10 in optimization are compared. Logarithmic scale is applied along the *y* coordinate.

**Figure 6 entropy-26-00522-f006:**
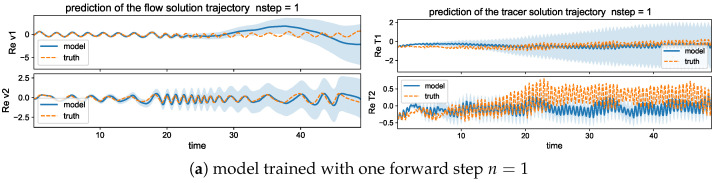
Trajectory prediction of flow and tracer solutions compared with the truth from direction model simulations. Performance of the trained models with different forward time steps n=1,10 are compared. The sample error variance is marked by the shared area around the prediction. The multistep training case n=10 has stable performance with tiny error variance.

**Figure 7 entropy-26-00522-f007:**
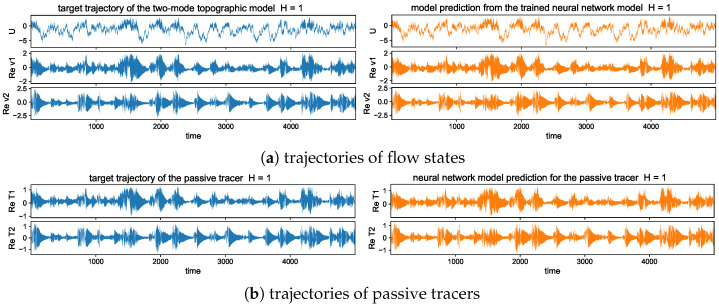
Long-time trajectory prediction of the flow and tracer solutions with weak topographic stress H=1 up to T=5000. The reference states from the direct model simulation (**left** panel) are compared with the model prediction (**right**) using the trained neural network model.

**Figure 8 entropy-26-00522-f008:**
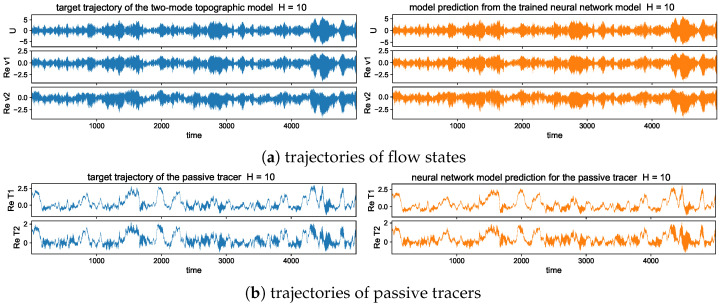
Long-time trajectory prediction of the flow and tracer solutions with strong topographic regime H=10 up to T=5000. The reference states from the direct model simulation (**left** panel) are compared with the model prediction (**right**) using the trained neural network model.

**Figure A1 entropy-26-00522-f0A1:**
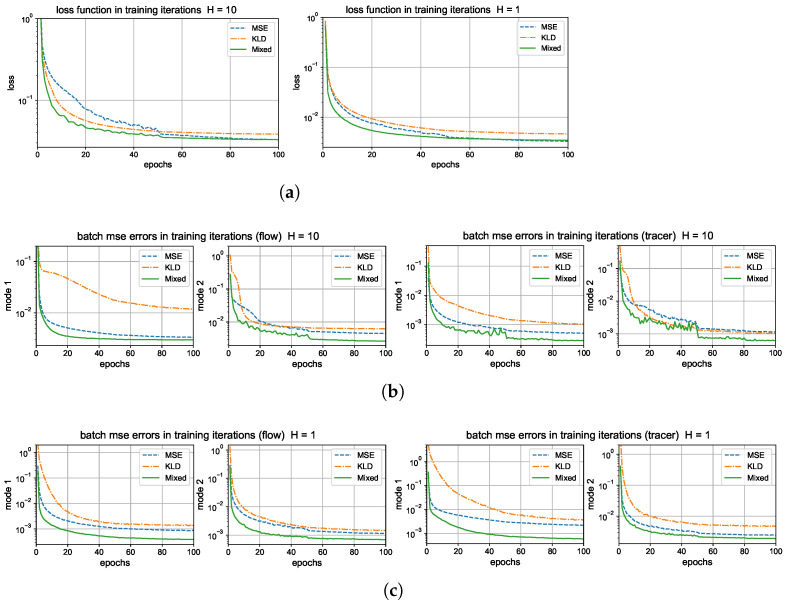
Training iterations of the full loss function value (**upper**) and training batch averaged mean square errors (**middle** and **lower**). Errors in each of the flow and tracer modes are compared under three different loss functions *L* for optimization: $L_{2}$ loss (MSE), KL divergence (KLD), and a mixed loss combining these two. Logarithmic scale is applied along the *y* coordinate. (**a**) Loss function with strong (H=10, **left**) and weak (H=1, **right**) topography. (**b**) Batch MSE for flow (**left**) and tracer (**right**) with strong topography H=10. (**c**) Batch MSE for flow (**left**) and tracer (**right**) with weak topography H=1.

**Figure A2 entropy-26-00522-f0A2:**
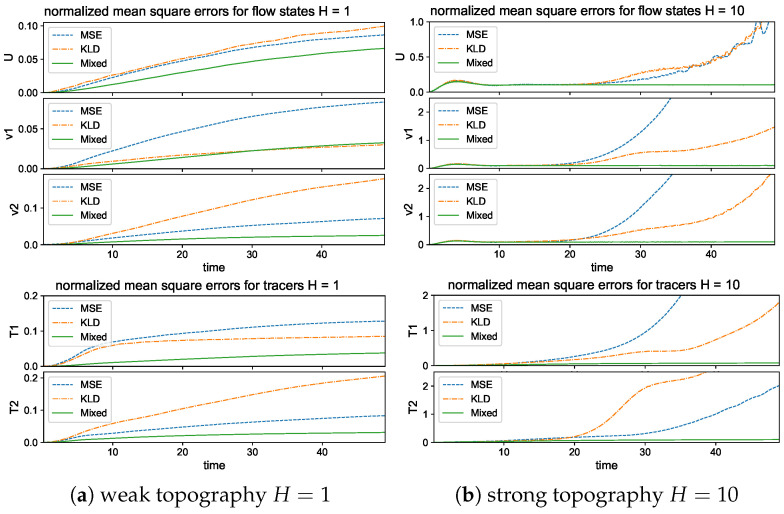
Comparison of prediction errors for models optimized with different loss functions in the two topography regimes H=1 and H=10. The normalized mean square errors are compared for both the flow states and tracer modes measured at recurrent predictions of 500 time steps up to T=50.

**Figure A3 entropy-26-00522-f0A3:**
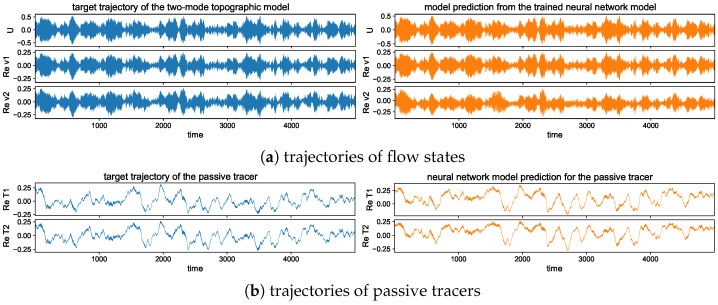
Long-time trajectory prediction of the flow and tracer solutions with weak white noise forcing $\sigma_{U}=\sigma_{0}$ in strong topographic regime H=10. The same trained model in the main text is used for this different forcing regime with distinct statistics.

**Figure A4 entropy-26-00522-f0A4:**
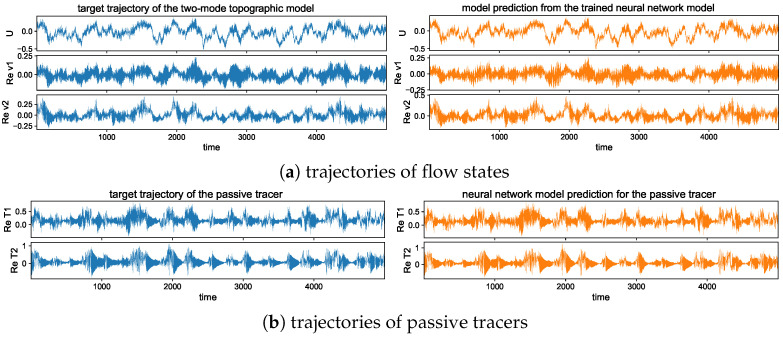
Long-time trajectory prediction of the flow and tracer solutions with weak white noise forcing $\sigma_{U}=\sigma_{0}$ in weak topographic regime H=1. The same trained model in the main text is used for this different forcing regime with distinct statistics.

**Figure A5 entropy-26-00522-f0A5:**
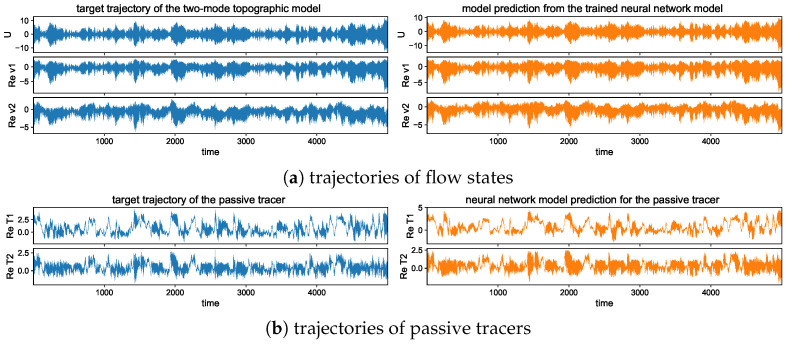
Long-time trajectory prediction of the flow and tracer solutions with strong white noise forcing $\sigma_{U}=20\sigma_{0}$ in strong topographic regime H=10. The same trained model in the main text is used for this different forcing regime with distinct statistics.

**Figure A6 entropy-26-00522-f0A6:**
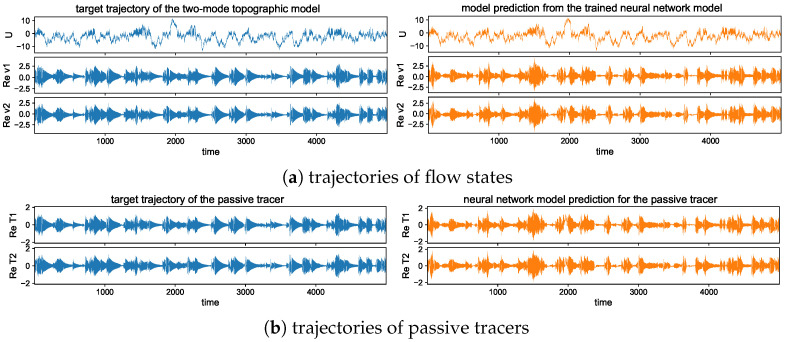
Long-time trajectory prediction of the flow and tracer solutions with strong white noise forcing $\sigma_{U}=20\sigma_{0}$ in weak topographic regime H=1. The same trained model in the main text is used for this different forcing regime with distinct statistics.

**Table 1 entropy-26-00522-t001:** Statistical error in mean (SME) and variance (SVE) from the ensemble prediction of the trained neural network model. The two parameter regimes with topographic stress H=1 and H=10 are compared. The same trained model is used for the statistical prediction with different white noise amplitudes $\sigma_{U}=10\sigma_{0},20\sigma_{0}$.

		$\sigma_{U}=10\sigma_{0}$				
		*U*	${\hat{v}}_{1}$	${\hat{v}}_{2}$	${\hat{T}}_{1}$	${\hat{T}}_{2}$
H=1	SME	3.03 × 10^−4^	2.49 × 10^−5^	5.53 × 10^−4^	1.96 × 10^−5^	7.60 × 10^−4^
	SVE	7.26 × 10^−3^	6.57 × 10^−2^	3.12 × 10^−2^	7.50 × 10^−2^	3.57 × 10^−2^
H=10	SME	5.34 × 10^−4^	2.87 × 10^−4^	9.16 × 10^−4^	4.29 × 10^−2^	8.05 × 10^−2^
	SVE	9.25 × 10^−1^	8.82 × 10^−2^	8.51 × 10^−2^	5.31 × 10^−2^	6.75 × 10^−2^
		$\sigma_{U}=20\sigma_{0}$				
		*U*	${\hat{v}}_{1}$	${\hat{v}}_{2}$	${\hat{T}}_{1}$	${\hat{T}}_{2}$
H=1	SME	7.98 × 10^−2^	8.50 × 10^−3^	2.03 × 10^−2^	6.18 × 10^−2^	1.92 × 10^−2^
	SVE	1.92 × 10^−1^	2.40 × 10^−1^	2.57 × 10^−1^	2.51 × 10^−1^	2.49 × 10^−1^
H=10	SME	2.84 × 10^−5^	6.77 × 10^−2^	2.53 × 10^−1^	4.60 × 10^−2^	1.44 × 10^−2^
	SVE	9.00 × 10^−1^	4.68 × 10^−1^	4.03 × 10^−1^	4.41 × 10^−1^	2.03 × 10^−1^

**Table A1 entropy-26-00522-t0A1:** Standard model hyperparameters for training the neural network model.

total training epochs	100
training batch size	100
starting learning rate	0.005
learning rate reduction rate	0.5
learning rate reduction at iteration step	50, 80
time step size between two measurements Δt	0.1
LSTM sequence length *m*	100
forward prediction steps in training *n*	10
hidden state size *h*	50
number of stages in LSTM cell *s*	4

## Data Availability

The data presented in this study are openly available in GitHub at https://github.com/qidigit/ (accessed on 30 April 2024).

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
