# Peer review of "Unambiguous Models and Machine Learning Strategies for Anomalous Extreme Events in Turbulent Dynamical System"

_entropy, 2024, doi:10.3390/e26060522_

Round 1

Reviewer 1 Report

Comments and Suggestions for Authors

The paper "Unambiguous Models and Machine Learning Strategies for Anomalous Extreme Events in Turbulent Dynamical Systems" introduces a novel neural network architecture designed to predict extreme events in turbulent systems. By incorporating a conditional Gaussian framework, the model overcomes limitations of traditional approaches, effectively learning and predicting key dynamical mechanisms and extreme events using limited data. Tested on a simplified geophysical turbulence model, the neural network demonstrates robust performance across various dynamical regimes, showing promise for applications in complex real-world turbulent systems. This work represents a significant advancement in the field of turbulence modeling by leveraging advanced neural network architectures. The paper is recommended for publication with optional minor revisions based on the following suggestions.

1. On page 10 of the draft, the author mentions: "We introduce a multi-stage structure in the one-step time update in each LSTM cell Lc(x, h)." Has the author tried to interpolate (linearly) the available data from large measurement time steps to finer ones and then train the single-stage neural network? Could the author comment on this?

2. In the draft, equations (2.2) describe a QG turbulence model, and by introducing spectral decomposition in (2.5), the author simplifies (2.2) into (2.6). In (2.2), the term $v\cdot \nabla q$ should involve the convolution among different modes when transferring into the spectral space; however, these terms are not present in (2.6). Is there something missing? Could the author provide further explanation?

Author Response

I would like to express my gratitude to the reviewer for her/his valuable comments and suggestions to improve the clarity and quality of this manuscript. The manuscript is revised according to the comments offered by the reviewer. Below we give a point-to-point reply according to the listed comments. New edits are marked with blue color in the revised manuscript to help expedite the rereading process.   1. On page 10 of the draft, the author mentions: "We introduce a multi-stage structure in the one-step time update in each LSTM cell Lc(x, h)." Has the author tried to interpolate (linearly) the available data from large measurement time steps to finer ones and then train the single-stage neural network? Could the author comment on this? Response: In the case with relatively large time step Δt in the data, it is usually not a good idea to interpolate the data and train it in a single-stage network. Especially in the case with frequent extreme events, this may miss the important extreme values in the training data and lead to wrong prediction result. In the topographic model tested here, using the single-stage neural network will usually lead to rapid divergence after only a few time iterations due to the strong internal instability due to the mean-fluctuation interactions. We emphasize this point at the end of the paragraph on page 10: In addition, using a single-stage update with the original LSTM will often lead to quick divergence in time iterations due to the inherent internal instability in the turbulent systems.   2. In the draft, equations (2.2) describe a QG turbulence model, and by introducing spectral decomposition in (2.5), the author simplifies (2.2) into (2.6). In (2.2), the term $v\cdot \nabla q$ should involve the convolution among different modes when transferring into the spectral space; however, these terms are not present in (2.6). Is there something missing? Could the author provide further explanation? Response: This is a very important point raised by the reviewer. Indeed, the fully nonlinear model will be much more complex involving all the nonlinear interactions between different scales. But this will also make the problem very complicated and untractable especially for the test of machine learning models. This is the inspiration we introduce the mean and fluctuation decomposition in (2.3), and focus in the first place on the nonlinear interactions only between the large-scale mean flow U and all the small-scale fluctuation states. In this case, we just assume the interactions among the fluctuation modes are much smaller than the mean-fluctuation interaction. In order to achieve this, we propose the so-called `layered topography' in (2.5). In this way, the nonlinear self-interactions between fluctuation modes will be automatically eliminated. Still, we agree when the turbulence of the system grows stronger, the nonlinear coupling between different scales will become more important and should not be neglected. This will be the focus of a next follow-up research and will require more additional treatment of the model. We added the new paragraph in Sec. 2.2 to emphasize this important point: Still, the model (2.2a) couples all the small-scale fluctuation modes through the nonlinear advection term on the left hand side. This will lead to very complex dynamical structures for analysis. Here, in order to find more analytically tractable solutions, we propose further simplification to the original system so that we are able to focus on the most important mean-fluctuation interactions in determining the final flow structure.

Reviewer 2 Report

Comments and Suggestions for Authors

Review of “Unambiguous Models and Machine Learning Strategies for Anomalous Extreme Events in Turbulent Dynamical System”

The author presents a study in which data-driven techniques are employed to learn the small-scale dynamics of a (simplified) turbulent flow problem. In particular, low-dimensional small-scale modes and passive tracer equations are derived from a barotropic flow model. Explicit expressions for the trajectories and statistical moments can be derived, which are used to inform the data-driven model. This model in turn is based on a modified LSTM architecture. Various loss functions for the LSTM are tested, and as well different training strategies involving a “multi-step” approach. The stability of the coupled system is examined, and the predictive peformance is tested for both trajectory matching as well as the accuracy of the long-term statistics. 

The article is interesting in my opinion, as it covers a lot of important topics. The results are also quite promising. I do find the text unclear at times, and I have asked for clarification though my comments below. These should be addressed before publication can be considered. 

It is not entirely clear to me what is meant with “unambiguous models”. My guess is that this refers to the 2 SDEs given by (2.1), but some more information could be useful about the general idea behind the framework. Does the conditional Gaussian p(u_2 | u_1) follow from this, and make clear how this Gaussian is relevant for the overall method that is proposed here.

(2.6a) (2.6b): some more information about how these equations are derived from (2.2a)(2.2b) should be given.

“ On the other hand, the above formulas with the conditional Gaussian structures imply that

the small-scale processes can be recovered from the statistics in the large-scale mean flow.” What formulas are referred to here? Also, can it be made more clear how the analytic results of this section (2.2) are used to inform the rest of the work?

The author talks about model error at several locations. What is meant by this? It is an ambiguous terms that can mean different things depending on the context.

The author modifies the LSTM by an inner iteration in the LSTM cells (the multi-stage LSTM cell), allowing for multiple time steps inside a larger time interval over which data is available. In the results section, are all LSTMs of the modified kind? If so, are the results markedly worse is no multi-stage approach is used, i.e. if a vanilla LSTM is employed? It would be good if the performance of this modification to the LSTM architecture is made clear. 

A “multi-step” training loss is also proposed. In my understanding this is separate from the “multi-stage” approach of the LSTM, but for clarity it would be good to emphasize this briefly, as the nomenclature is very similar.  Also, am I to understand that the multi-step training is performed “online”, i.e. is the LSTM trained while coupled to the dynamical system? If so this should ebe made clear. 

Some more minor comments:

(2.2b): psi’ is not defined

Section 5.2, page 17: small change in notation, the loss is summed to n, wheras in (4.6) it is summed to (capital) N 

Author Response

I would like to thank the reviewer for the many valuable comments and suggestions to improve the clarity and quality of this manuscript. The manuscript is revised according to the comments offered by the reviewer. Below we give a point-to-point reply according to the listed comments. New edits are marked with blue color to help expedite the rereading process. We hope that we have been able to address all the concerns raised from the reviewer.   It is not entirely clear to me what is meant with “unambiguous models”. My guess is that this refers to the 2 SDEs given by (2.1), but some more information could be useful about the general idea behind the framework. Does the conditional Gaussian p(u_2 | u_1) follow from this, and make clear how this Gaussian is relevant for the overall method that is proposed here. Response: We thank the reviewer for this important clarification question. The unambiguous model refers to using the conditional Gaussian framework (2.1) for the analysis of the multiscale turbulent systems (2.2) especially including extreme events. The conditional Gaussian distribution is derived based on this conditional framework. The neural network models are based on this conditional Gaussian structure and are developed through this particular structures for the large number of unresolved small scales. We realize we were not very clear at this point in constructing the model. Additional explanations are added in the introduction section to clarify the main ideas.   (2.6a) (2.6b): some more information about how these equations are derived from (2.2a)(2.2b) should be given. Response: More explanation for each terms in (2.6) are added according to the layered topography structure assumed to the model and how each term is reached under the layered spectral projection.   “ On the other hand, the above formulas with the conditional Gaussian structures imply that the small-scale processes can be recovered from the statistics in the large-scale mean flow.” What formulas are referred to here? Also, can it be made more clear how the analytic results of this section (2.2) are used to inform the rest of the work? Response: We added the explicit references to the mean and variance equation numbers (2.9)-(2.11). The analytic solutions are first used to imply the key statistical structures of the mean and fluctuation coupling mechanisms in the topographic model. From the formulas, it explicitly shows the dependence of the large-scale mean flow in the small-scale dynamics. This sets up the foundation for the development of the machine learning models based on the physics model in large scales. Only through these explicit structures we are able to use the neural network models next to learn such unresolved small-scale processes. In addition, the explicit formulas will also assist identifying the different dynamical regimes where extreme events are developed and offer general guideline for the design of neural network architectures. More clarifications about the link between the analytic equations and the machine learning methods are added at the end of section 2.2 as well as in the introduction section.   The author talks about model error at several locations. What is meant by this? It is an ambiguous terms that can mean different things depending on the context. Response: We thank the reviewer for pointing out this ambiguity. More specific definition on what exactly the model error is referred each time it is mentioned.   The author modifies the LSTM by an inner iteration in the LSTM cells (the multi-stage LSTM cell), allowing for multiple time steps inside a larger time interval over which data is available. In the results section, are all LSTMs of the modified kind? If so, are the results markedly worse is no multi-stage approach is used, i.e. if a vanilla LSTM is employed? It would be good if the performance of this modification to the LSTM architecture is made clear. Response: It is found that a vanilla LSTM is usually not valid in the trajectory prediction of stochastic signals in turbulent systems. Especially in the tested cases with frequent extreme events, this may miss the important extreme values in the training data and cannot give reasonable prediction result. Using the single-stage neural network will usually lead to rapid divergence after only a few time iterations due to the strong internal instability due to the mean-fluctuation interactions. Similar divergence results are shown and discussed in [27], so we choose to focus on the new multi-stage model in this paper. We emphasize this point more clearly at the end of the paragraph on page 10.   A “multi-step” training loss is also proposed. In my understanding this is separate from the “multi-stage” approach of the LSTM, but for clarity it would be good to emphasize this briefly, as the nomenclature is very similar. Also, am I to understand that the multi-step training is performed “online”, i.e. is the LSTM trained while coupled to the dynamical system? If so this should be made clear. Response: We thank the reviewer for pointing out this confusion in notations. The reviewer is correct the LSTM is trained with coupling to the physical model. More clarifications are added on pages 15-16.   Some more minor comments: (2.2b): psi’ is not defined Section 5.2, page 17: small change in notation, the loss is summed to n, wheras in (4.6) it is summed to (capital) N Response: The definition is added and typos have been corrected.

Reviewer 3 Report

Comments and Suggestions for Authors

Dear Editor,

Please find attached my comments.

Author Response

I would like to express my gratitude to the reviewer for her/his valuable comments and suggestions to improve the clarity and quality of this manuscript. The manuscript is revised according to the comments offered by the reviewer. Below we give a point-to-point reply according to the listed comments. New edits are marked with blue color to help expedite the rereading process. We hope that we have been able to address all the concerns raised from the reviewer.   • How do you explain the use of the Gaussian process as a conditional posterior probability (formula after (2.1)? This assumption is pretty restrictive and normally not the case of any realistic process. I understand that its a convenient choice that gives you better results, however, why not to use the general exponential family of pdf containing more pdfs, also the heavy-tailed ones? Response: This is a very important question. Indeed, the conditional Gaussian structures may not be always valid for realistic systems and could be subject to additional assumptions. We adopt this framework first since it fits well to the topographic models studied here. Also, it can be shown in many more general applications, even though not strictly satisfying the conditional Gaussian property, the equations (2.1) can be still used as an imperfect approximation with model errors. The reviewer's suggestion of using more general family of exponential family of PDFs is also a great idea. We are trying some applications with this group of PDFs as a followup research. We added more clarification on using this conditional structure on page 4.   • In Sec.3.2 I find the use of different loss functions an interesting idea. How do you generally select the α parameter in (3.7c)? Response: This is also a very interesting question by the reviewer. At the current stage, we just pick the parameter α empirically. The reason is only to introduce a balance factor between the pointwise L2 loss and the KL-divergence for measuring the distribution. Indeed it will be very interesting to carry out a systematic investigation for the role of different values of α next. We added clarification below (3.7c) for the choice of the parameter.   • I did not understand what type of heavy-tailed pdfs arise in the shown simulations? The mixture of Gaussian pdfs can’t give a heavy tailed pdf and can’t be used to model its tail. Response: It is correct that we cannot use Gaussian PDFs to model such heavy tails shown in Fig. 2.2. In fact, these heavy tails are due to the mixture of conditional Gaussian distributions conditional on different realizations of the mean state U. That is, these are the summation of PDFs conditional on the random process U, thus an additional average about U leads to the non-Gaussian distributions. We added more explanations in the last paragraph on page 8 to clarify this point.